# Long-term Evaluation of Commercial Air Quality Sensors: An Overview from the QUANT Study

Sebastian Diez[1,2], Stuart Lacy[2], Hugh Coe[3], Josefina Urquiza[4,5], Max Priestman[6], Michael Flynn[3], Nicholas Marsden[3], Nicholas A. Martin[7], Stefan Gillott[6], Thomas Bannan[3], Pete Edwards[2]

[1]Centro de Investigación en Tecnologías para la Sociedad, Universidad del Desarrollo, Santiago, Chile, CP 7550000

[2]Wolfson Atmospheric Chemistry Laboratories, University of York, York, YO10 5DD, UK

[3]Department of Earth and Environmental Science, Centre for Atmospheric Science, School of Natural Sciences, The University of Manchester, Manchester, M13 9PL, UK

[4]Grupo de Estudios de la Atmósfera y el Ambiente (GEAA), Universidad Tecnológica Nacional, Facultad Regional Mendoza (UTN-FRM), Cnel. Rodriguez 273, Mendoza, 5501, Argentina

[5]Consejo Nacional de Investigaciones Científicas y Técnicas (CONICET) Argentina

[6]MRC Centre for Environment and Health, Environmental Research Group, Imperial College, London, W12 0BZ, UK

[7]National Physical Laboratory, Teddington TW11 0LW, UK

*Correspondence:* Sebastian Diez (sebastian.diez@udd.cl); Pete Edwards (pete.edwards@york.ac.uk)

**Abstract.** In times of growing concern about the impacts of air pollution across the globe, lower-cost sensor technology is giving the first steps in helping to enhance our understanding and ability to manage air quality issues, particularly in regions without established monitoring networks. While the benefits of greater spatial coverage and real-time measurements that these systems offer are evident, challenges still need to be addressed regarding sensor reliability and data quality. Given the limitations imposed by intellectual property, commercial implementations are often "black boxes", which represents an extra challenge as it limits end-users' understanding of the data production process. In this paper we present an overview of the QUANT (Quantification of Utility of Atmospheric Network Technologies) study, a comprehensive 3-year assessment across a range of urban environments in the United Kingdom, evaluating 43 sensor devices, including 119 gas sensors and 118 particulate matter sensors, from multiple companies. QUANT stands out as one of the most comprehensive studies of commercial air quality sensor systems carried out to date, encompassing a wide variety of companies in a single evaluation and including two generations of sensor technologies. Integrated into an extensive data set open to the public, it was designed to provide a long-term evaluation of the precision, accuracy, and stability of commercially available sensor systems. To attain a nuanced understanding of sensor performance, we have complemented commonly used single-value metrics (e.g., Coefficient of Determination ($R^2$), Root Mean Square Error (RMSE), Mean Absolute Error (MAE)) with visual tools. These include Regression plots, Relative Expanded Uncertainty (REU) plots, and Target plots, enhancing our analysis beyond traditional metrics. This overview discusses the assessment methodology, and key findings showcasing the significance of the study. While more comprehensive analyses are reserved for future detailed publications, the results

shown here highlight the significant variation between systems, the incidence of corrections made by manufacturers, the effects of relocation to different environments, and the long-term behaviour of the systems. Additionally, the importance of accounting for uncertainties associated with reference instruments in sensor evaluations is emphasised. Practical considerations in the application of these sensors in real-world scenarios are also discussed, and potential solutions to end-user data challenges are presented. Offering key information about the sensor systems' capabilities, the QUANT study will serve as a valuable resource for those seeking to implement commercial solutions as complementary tools to tackle air pollution.

**Keywords:** air pollution, commercial sensor systems, QUANT, long-term evaluation.

## 1. Introduction

Emerging lower-cost sensor systems[1] offer a promising alternative to the more expensive and complex monitoring equipment traditionally used for measuring air pollutants such as $PM_{2.5}$, $NO_2$, and $O_3$ (Okure et al., 2022). These innovative devices hold the potential to expand spatial coverage (Malings et al., 2020) and deliver real-time air pollution measurements (Tanzer-Gruener et al., 2020). However, concerns regarding the variable quality of the data they provide still hinder their acceptance as reliable measurement technologies (Karagulian et al., 2019; Zamora et al., 2020).

Sensors[2] face key challenges such as cross-sensitivities (Bittner et al., 2022; Cross et al., 2017; Levy Zamora et al., 2022; Pang et al., 2018), internal consistency (Feenstra et al., 2019; Ripoll et al., 2019), signal drift (A. Miech et al., 2023; Li et al., 2021; Sayahi et al., 2019), long term performance (Bulot et al., 2019; Liu et al., 2020) and data coverage (Brown & Martin, 2023; Duvall et al., 2021; Feinberg et al., 2018). Additionally, environmental factors such as temperature and humidity (Bittner et al., 2022; Farquhar et al., 2021; Crilley et al., 2018; Williams, 2020) can significantly influence sensor signals.

In recent years, manufacturers of both sensing elements (Han et al., 2021; Nazemi et al., 2019) and sensor systems have made significant technological advances (Chojer et al., 2020). For example, there are now commercial and non-commercial systems equipped with multiple detectors to measure distinct pollutants (Buehler et al., 2021; Hagan et al., 2019; Pang et al., 2021) helping to mitigate the effects of cross-interferences. Additionally, enhancements in electrochemical OEMs have been demonstrated in terms of their specificity (Baron & Saffell, 2017; Ouyang, 2020).

However, the complex nature of their responses, coupled with their dependence on local conditions means sensor performance can be inconsistent (Bi et al., 2020). This complicates the comparison of results or anticipating sensor future performance across different studies. Moreover, assessments of sensor performance found in the academic

---

[1] The term "sensor systems" refers to sensors housed within a protective case, which includes a sampling and power system, electronic hardware and software for data acquisition, analog-to-digital conversion, data processing and their transfer (Karagulian et al., 2019). Unless specified otherwise, the term "sensor" will be used as a synonym of "sensor systems". Other alternative names for "sensor systems" used here are "sensor devices" (or "devices"), "sensor units" (or "units").

[2] In a narrower sense, "sensor" typically denotes the specific component within a sensor system that detects and responds to environmental inputs, producing a corresponding output signal. To distinguish this from the broader use of "sensor" as equivalent to "sensor system" in our text, we will utilise alternative terms such as "detector", "sensing element", or "OEM" (original equipment manufacturer) when referring specifically to this component, thereby preventing confusion.

literature often rely on a range of protocols (e.g., CEN (2021) and Duvall et al. (2021)) and data quality metrics (e.g.,
Spinelle et al. (2017) and Zimmerman et al. (2018)), with many studies limited to a single-site co-location and/or
short-term evaluations that do not fully account for broader environmental variations (Karagulian et al., 2019).
The calibration of any instrument used to measure atmospheric composition is fundamental to guarantee their accuracy
(Alam et al., 2020; Long et al., 2021; Wu et al., 2022). Using out-of-the-box sensor data without fit-for-purpose
calibration can produce misleading results (Liang & Daniels, 2022). An effective calibration not only involves
identifying but also compensating for estimated systematic effects in the sensor readings, a process defined as a
correction (for a detailed definition and differentiation of calibration and correction see JCGM, 2012). For standard
air pollution measurement techniques, calibration is often performed in a controlled laboratory environment (Liang,
2021). For example, for gases, a known concentration is sampled from a certified standard. Similarly, for PM, particles
of known density and size are generated. Both gases and PM calibration are conducted under controlled airflow
conditions
Yet, the aforementioned challenges with lower-cost sensor-based devices suggest that such calibrations may not
always accurately reflect real-world conditions (Giordano et al., 2021). A frequent approach involves co-locating
sensors alongside regulatory instruments in their intended deployment areas and/or conditions and using data-driven
methods to match the reference data (Liang & Daniels, 2022). Numerous studies have investigated the effectiveness
of calibration methods for sensors e.g. (Bigi et al., 2018; Bittner et al., 2022; Malings et al., 2020; Spinelle et al., 2017;
Zimmerman et al., 2018), including selecting appropriate reference instruments (Kelly et al., 2017), the need for
regular calibration to maintain accuracy (Gamboa et al., 2023), the necessity of rigorous calibration protocols to ensure
consistency (Kang et al., 2022), and transferability (Nowack et al., 2021) of results. Ultimately, the reliability and
associated uncertainty of any applied calibration will influence the final sensor data quality.
For end-users to make informed decisions on the applicability of air pollution sensors, a realistic understanding of the
expected performance in their chosen application is necessary (Rai et al., 2017). Despite this, there has been relatively
little progress in clarifying the performance of sensors for air pollution measurements outside of the academic arena.
This is largely due to the significant variability in both the number of sensors and the variety of applications tested,
compounded by the proliferation of commercially available sensors/sensor systems with different configurations.
Furthermore, the access to highly accurate measurement instrumentation and/or regulatory networks remains limited
for those outside of the atmospheric measurement academic field (e.g. Lewis and Edwards (2016) and Popoola et al.
(2018)). From a UK clean air perspective, this ambiguity represents a major problem. The lack of a consistent message
undermines the exploitation of these devices' unique strengths, notably their capability to form spatially dense
networks with rapid time resolution. Consequently, there is potential for a mismatch in users' expectations of what
sensor systems can deliver and their actual operating characteristics, eroding trust and reliability.
In this work, as part of the UK Clean Air program funded QUANT project, we deployed a variety of sensor
technologies (43 commercial devices, 119 gas and 118 PM measurements) at 3 representative UK urban sites —
Manchester, London and York— alongside extensive reference measurements, to generate the data for an
comprehensive in-depth performance assessment. This project aims to not only evaluate the performance of sensor
devices in a UK urban climatological context but also provide critical information for the successful application of
these technologies in various environmental settings. To our knowledge, QUANT is the most extensive and longest-
running evaluation of commercial sensor systems globally to date. Furthermore, we tested multiple manufacturers'
data products, such as out-of-the-box data versus locally calibrated data, for a significant number of these sensors to

understand the implications of local calibration. This comprehensive approach offers unprecedented insights into the operational capabilities and limitations of these sensors in real-world conditions. Significantly, some of the insights gathered during QUANT have contributed to the development of the Publicly Available Specification (PAS 4023, 2023), which provides guidelines for the selection, deployment, maintenance, and quality assurance of air quality sensor systems. While this manuscript serves as an initial overview, detailed analyses of the measured pollutants and study phases, offering a more comprehensive perspective on sensor performance, are planned for future publications.

In the following sections, we delve into the methodology and provide an overview of the QUANT dataset, as well as a discussion of some of the key findings and potential considerations for end-users.

## 2. QUANT study design

To capture the variability of UK urban environments, identical units were installed at three carefully selected field sites. Two of these sites are highly instrumented urban background measurement supersites: the London Air Quality Supersite (LAQS; for more details, refer here: https://uk-air.defra.gov.uk/networks/site-info?site_id=HP1) and the Manchester Air Quality Supersite (MAQS; for more details, see: http://www.cas.manchester.ac.uk/restools/firs/), located in densely populated urban areas with unique air quality challenges. The third site is a roadside monitoring site in York, which is part of the Automatic Urban and Rural Network (AURN; click here for more details: https://uk-air.defra.gov.uk/networks/site-info?uka_id=UKA00524&search=View+Site+Information&action=site&provider=archive), representing a urban environment more influenced by traffic. This selection strategy ensures that the QUANT study's findings reflect the dynamics of urban air quality across different UK settings, while providing comprehensive reference measurements. Further details about each site can be found in Section S1 in the Supp.

### 2.1 Main study

The Main QUANT assessment study aimed to perform a transparent long-term (19 Dec 2019 - 31 Oct 2022) evaluation of commercially available sensor technologies for outdoor air pollution monitoring in UK urban environments. Four units of five different commercial sensor devices (Table 1) were purchased in Sept 2019 for inclusion in the study, with the selection criteria being: market penetration and/or previous performance reported in the literature, ability to measure pollutants of interest (e.g. $NO_2$, $NO$, $O_3$, and $PM_{2.5}$), and capacity to run continuously reporting high time resolution data (1-15 min data) ideally in near real-time (i.e., available within minutes of measurement) with data accessible via an API.

**Table 1. Main QUANT devices description. The 20 units, all commercially available and ready for use as-is, offered 56 gas and 56 PM measurements in total. For a detailed description of the devices see Section S3 in the Supp.**

| Product* (# units) | Company[3] | Measurements | | | | | | | | Cost (£)** |
|---|---|---|---|---|---|---|---|---|---|---|
| | | NO | NO$_2$ | O$_3$ | CO | CO$_2$ | PM$_1$ | PM$_{2.5}$ | PM$_{10}$ | |
| AQY (4) | Aeroqual | - | ✓ | ✓ | - | - | - | ✓ | ✓ | ~4.7K |

---

[3] Throughout this article, the terms "manufacturers" and "company" are used interchangeably to refer to entities that produce, and/or sell sensor systems or devices. This usage reflects the industry practice of referring to businesses involved in the production and distribution of technology products without distinguishing between their roles in manufacturing or sales.

| AQM (4) | AQMesh | ✓ | ✓ | ✓ | - | ✓ | ✓ | ✓ | ✓ | ~8.6K |
| Ari (4) | QuantAQ | ✓ | ✓ | ✓ | ✓ | ✓ | ✓ | ✓ | ✓ | ~8.6K |
| PA (4) | PurpleAir | - | - | - | - | - | ✓ | ✓ | ✓ | ~0.3K |
| Zep (4) | Earthsense | ✓ | ✓ | ✓ | - | - | ✓ | ✓ | ✓ | ~7K |

*AQY: Aeroqual; AQM: AQMesh; Ari: Arisense; PA: PurpleAir; Zep: Zephyr. **Cost (Sep 2019) per unit including UK taxes and associated contractual costs (i.e., communication, data access, sensor replacement, etc.).

Initially, all the sensors were deployed in Manchester for approximately 3 months (mid-Dec 2019 to mid-Mar 2020) before being split up amongst the three sites (Fig. 1). At least one unit per brand was re-deployed to the other two sites (mid-March 2020 to early-July 2022) leaving two devices per company in Manchester to assess inter-device consistency. In the final 4 months of the study, all the sensor systems were relocated back to Manchester (early July 2022 to the end of October 2022).

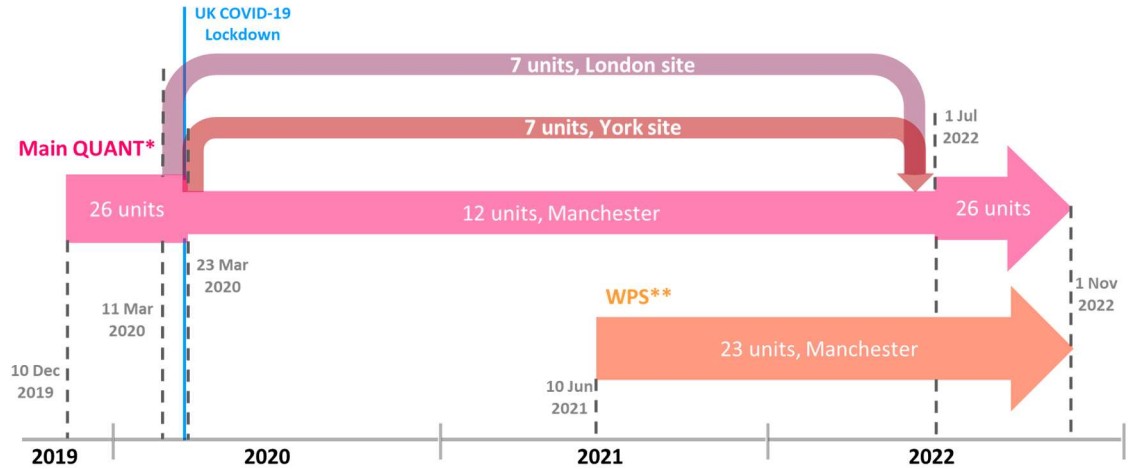

*: Aeroqual (x4), AQMesh (x4), Zephyr (x4), QuantAQ (x4), PurpleAir (x10)

**: AQMesh (x3), Bosch (x2), Clarity (x3), Kunak (x3), Oizom (x2), QuantAQ (x3), South Coast Science (x2), Respirer Living Sciences (x2), Vortex (x3)

**Figure 1. Main QUANT and Wider Participation Study (WPS) timeline.**

## 2.2 Wider Participation Study

The Wider Participation Study (WPS) was a no-cost complementary extension of the QUANT assessment, specifically designed to foster innovation within the air pollution sensors domain. This segment of the study took place entirely at the MAQS from 10th June 2021 to 31st October 2022 (Fig. 1). It included a wider array of commercial platforms (9 different sensor systems brands), and offered manufacturers the opportunity to engage in a free-of-charge impartial evaluation process. Although participation criteria matched those of the Main QUANT study, a key distinction lay in the voluntary nature of participation: manufacturers were invited to contribute multiple sensor devices throughout the WPS study (see Table 2). Participants were able to demonstrate their systems' performance against collocated high-resolution (1-minute) reference data at a state-of-the-art measurement site such as the Manchester supersite.

**Table 2. The 23 WPS devices deployed at the Manchester supersite, all commercially available and ready for use as-is, provided 63 gases and 62 PM measurements in total. For a detailed description of the devices see the Section S4 in the Supp.**

| Product* (# units) | Company | Measurements | | | | | | | |
|---|---|---|---|---|---|---|---|---|---|
| | | NO | $NO_2$ | $O_3$ | CO | $CO_2$ | $PM_1$ | $PM_{2.5}$ | $PM_{10}$ |
| Mod (3) | QuantAQ | - | - | - | - | - | ✓ | ✓ | ✓ |
| AQM (3) | AQMesh | ✓ | ✓ | ✓ | ✓ | ✓ | ✓ | ✓ | ✓ |
| Atm (2) | RLS** | - | - | - | - | - | ✓ | ✓ | ✓ |
| IMB (2) | Bosch | - | ✓ | ✓ | - | - | - | ✓ | ✓ |
| Poll (2) | Oizom | ✓ | ✓ | ✓ | ✓ | ✓ | - | ✓ | ✓ |
| AP (3) | Kunak | ✓ | ✓ | ✓ | ✓ | ✓ | ✓ | ✓ | ✓ |
| SA (3) | Vortex IoT | - | ✓ | ✓ | - | - | - | ✓ | ✓ |
| NS (3) | Clarity | - | ✓ | - | - | - | ✓ | ✓ | ✓ |
| Prax (2) | SCS*** | ✓ | ✓ | ✓ | ✓ | ✓ | ✓ | ✓ | ✓ |

*Mod: Modulair; AQM: AQMesh; Atm: Atmos, Poll: Polludrone; AP: Kunak Air Pro; SA: Silax Air, NS: Node-S, Prax: Praxis.

**RLS: Respirer Living Sciences. ***SCS: South Coast Science.

## 2.3 Sensor deployment and data collection

All sensor devices were installed at the measurement sites as per manufacturer recommendations, adhering strictly to manufacturers' guidelines for electrical setup, mounting, cleaning, and maintenance guaranteed proper installation. Since all deployed systems were designed for outdoor use, no additional protective measures were necessary. Each of the systems were mounted on poles acquired specifically for the project or on rails at the co-location sites, without the need for special protections. Following the manufacturer's suggestions, sensors were positioned within 3 metres of the reference instruments' inlets. Custom electrical setups were developed for each sensor type, incorporating local energy sources and weather-resistant safety features, alongside security measures to deter vandalism and ensure uninterrupted operation. Routine maintenance was conducted monthly, although the COVID-19 pandemic necessitated longer intervals between visits. Despite these obstacles, efforts to maintain sensor security and functionality continued unabated, employing both physical safeguards and remote monitoring to preserve data integrity.

In addition to the device supplier's own cloud storage (accessed on-demand via each supplier's web portals), an automated daily scraping of each company's API was performed to save data onto a secure server at the University of York to ensure data integrity. Unlike other brands that utilise mobile data connections, PurpleAir sensors rely on WiFi for data transmission. Due to poor internet signal at the sites, we locally collected and manually uploaded readings for these units. Minor pre-processing was applied at this stage, including temporal harmonisation to ensure that all measurements had a minimum sampling period of 1-minute, ensuring consistency in measurement units and labels, and coercing into the same format to allow for full compatibility across sensor units. No additional modifications to the original measurements were applied; missing values were kept as missing and no additional flags were created based on the measurements beyond those provided by the manufacturers. For an overview of the sensor measurands and their corresponding data time resolutions as provided by the companies participating in the Main QUANT study and the WPS, please see Seccion S3 and S4 (Table S4 and S5) respectively.

## 2.4 Data products and co-located reference data

In addition to providing an independent assessment of sensor performance, QUANT also aimed to contribute to device manufacturers to help advance the field of air pollution sensors. During QUANT, device calibrations were performed solely at the discretion of the manufacturers without any intervention from our team, thus limiting the involvement of manufacturers in the provision of standard sensor outputs and unit maintenance as would be required by any standard customer. This approach enabled manufacturers to independently assess and benchmark their sensors' performance, using provided reference data to potentially develop calibrated data products. It's noteworthy that not all manufacturers chose to utilise these data for corrections or enhancements. However, those who did were expected to create and submit calibrated data products, subsequently named as "out-of-box" (initial data product), "cal1" (first calibrated product), and "cal2" (second calibrated product). This differentiation highlighted the varying degrees of engagement and application of the reference data by different manufacturers. Figures S2 and S3 (section S3 and S4 respectively) show a time-line of the different data products.

To this end, three separate 1-month periods of reference data, spaced every 6 months, were shared with each supplier, provisional data soon after each period, and ratified data when available. All reference data were embargoed until it was released to all manufacturers simultaneously to ensure consistency across manufacturers. For an overview of reference and equivalent-to-reference instrumentation, as defined in the European Union Air Quality Directive 2008/50/EC (hereafter referred to as EU AQ Directive), at each site, please refer to Section S2 (Table S1). For details on the quality assurance procedures applied to the reference instruments, see Table S2. To see the dates and periods of the shared reference data refer to Table S3.

**3. Results and discussion**

A key challenge in sensor performance evaluation is the high spatial and temporal variability errors that impact the accuracy of their readings, making the application of laboratory corrections more challenging. Furthermore, the overreliance on global performance metrics is a significant concern in sensor assessment. The Coefficient of Determination ($R^2$), Root Mean Squared Error (RMSE), and Mean Absolute Error (MAE) are among the most popular single-value metrics for evaluating sensor performance, alongside others (e.g., the bias, the slope and intercept of the regression fit). However, while single-value metrics offer an overview of performance, they can be limiting or misleading. They condense vast amounts of data into a single value, simplifying complexity at the expense of a nuanced understanding of error structures and information content (Diez et al., 2022), potentially overlooking critical aspects of sensor performance (Chai & Draxler, 2014). Visualisation tools (such as Regression plots, Target plots, and Relative Expanded Uncertainty plots) complement these metrics, allowing end users to identify relevant features, which could be beyond the scope of global metrics. For additional details on the metrics utilised in this study, including some of their limitations and advantages refer to section "S5. Performance Metrics". This section also provides a summary of current guidelines and standardisation initiatives, which may offer a foundation for end-users to select appropriate metrics for their own analyses (refer to table S6). For further discussion on metrics and visualisation tools for performance evaluation, readers are directed to Diez et al. (2022).

In response to these challenges, the QUANT assessment represents the most extensive independent appraisal of air pollution sensors in UK urban atmospheres. As the results presented here illustrate, QUANT is dedicated to examining sensor performance through multiple complementary metrics and visualisation tools, aiming to integrate these to accurately reflect the complexity of this dataset. This methodology promotes a nuanced understanding of sensor performance, extending beyond the limitations of conventional global single-value metrics.

Furthermore, by providing open access to the dataset, we encourage stakeholders to explore and utilise the data according to their unique needs and contexts, as detailed in the "Data Availability" section. In addition, we have developed a publicly accessible analysis platform (https://shiny.york.ac.uk/quant/), designed for straightforward offline analysis of the QUANT dataset. This platform enables users to interactively visualise the data through various representations, such as time series, regression plots, and Bland-Altman plots. It also offers statistical parameters (including regression equation, $R^2$, and RMSE) for analysing different pollutants, selecting specific sensors or manufacturers, and comparing across various co-location timeframes.

The following sections aim to provide an overview of the data and provide initial findings, with a focus on those that are most relevant to end-users of these technologies. The majority of examples presented here focus on $PM_{2.5}$ and $NO_2$ measurements, due to both a larger dataset available for these pollutants and their critical role in addressing the exceedances that predominantly impact UK air quality. All metrics and plots presented here are based on 1-hour averaged data. Unless otherwise specified, a data inclusion criterion of 75% was uniformly applied across our analyses to ensure the reliability and representativeness of the results. This threshold aligns with the EU AQ Directive, which mandates this proportion when aggregating air quality data and calculating statistical parameters. To highlight broad implications and insights into sensor technology, rather than focusing on the performance of specific manufacturers, figures illustrating brand-specific features have been anonymized. This is intended to prevent potential bias and encourage a holistic view of the data, ensuring interpretations remain focused on general trends rather than isolated examples.

### 3.1 Inter-device precision

Inter-device precision refers to the consistency of measurements across multiple identical devices (i.e., same brand and model), an important characteristic to ensure the reliability of sensor outputs over time (Moreno-Rangel et al., 2018). During QUANT, all the devices were collocated for the first 3 months and the final 3 months of the deployment to assess inter-device precision and its changes over time. Fig. 2 shows the inter-device precision (as defined by the CEN/TS 17660-1:2021, i.e., the "between sensor system uncertainty" metric: $u_s(b_s, s)$) of $PM_{2.5}$ measurements during these periods. For an overview of $NO_2$ and $O_3$ inter-device precision, see the "S6. Complementary plots" section in the supplementary (figures S4 and S5). While most of the companies display a certain level of inter-device precision stability in each period (except for one, with a seemingly upward trend in the final period), there are evident long-term changes. Notably, out of the four manufacturers assessed in the final period (each having 3 devices running simultaneously), three experienced a decline in their inter-device precision compared to two years earlier. This is likely due to both hardware degradation but also drift in the calibration, which at this point had been applied between 16 and 34 months prior (depending on the manufacturer). For extended periods, inconsistencies among devices from the same manufacturer might emerge, leading to varying readings under similar conditions. Consequently, data collected from different devices may not be directly comparable, which could result in inaccuracies or misinterpretations when analysing air quality trends or making decisions.

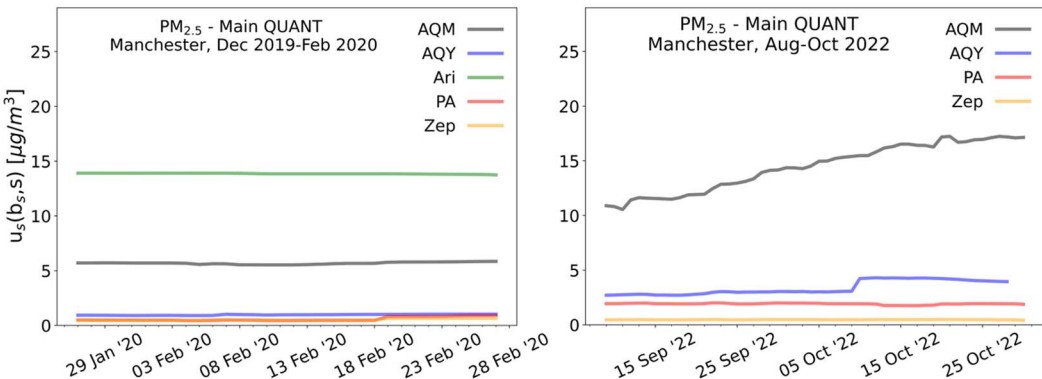

251

**Figure 2. The inter-device precision of PM$_{2.5}$ measurements from "identical" devices across the 5 companies participating in QUANT is assessed using the "between sensor system uncertainty" metric (defined by the CEN/TS 17660-1:2021 as *u(bs, s)*). Each line represents this metric as a composite of all sensors per brand (excluding units with less than 75% data) within a 40-day sliding window.**

It is worth noting that the inter-device precision provides no information on the accuracy of the sensor measurements; a batch of devices may provide a highly consistent, but also highly inaccurate measurement of the target pollutant.

The "target plot" (as shown in Fig. 3) is a tool commonly used to depict the bias/variance decomposition of an instrument's error relative to a reference (for more details see Jolliff et al. (2009)). The mean bias error (MBE) is used to characterise accuracy and precision is quantified by the centered Root Mean Squared Error (cRMSE, e.g. Kim et al. (2022) also called unbiased Root Mean Squared Error (uRMSE, e.g. Guimarães et al. (2018))). Fig. 3 visualises the performance of a set of PM$_{2.5}$ sensors of the WPS deployment for the first 2 months (out-of-box data) and the last 3 months of colocation (manufacturer-supplied calibrations). In addition to showcasing inter-device precision, Fig. 3 also serves as a transition to accuracy evaluation (the focus of the subsequent section).

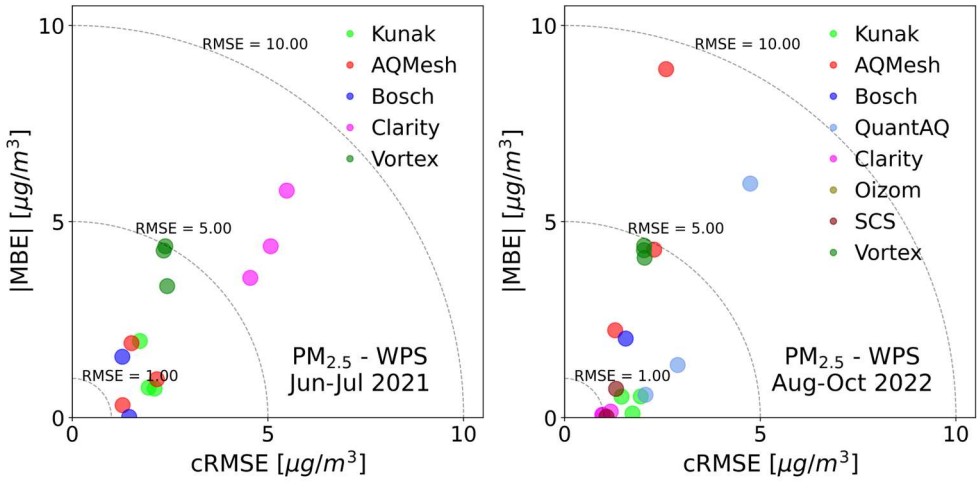

**Figure 3. Target diagrams for the WPS PM$_{2.5}$ measurements during the initial co-location period (Jun-Jul 2021, left) and final co-location period (Aug-Oct 2022, right). The error (RMSE) for each instrument is decomposed into the MBE (y-axis) and cRMSE (x-axis). Each point represents an individual sensor device, with duplicate devices having the same colour. Since only units with more than 75% of the data were considered, the plot on the right shows fewer units than the plot on the left.**

### 3.2 Device accuracy and co-location calibrations

Sensor measurement accuracy denotes how close a sensor's readings are to reference values (Wang et al., 2015). Characterising this feature is imperative for establishing sensor reliability and making informed decisions based on its data. Fig. 4 shows that co-location calibration can greatly impact observed $NO_2$ sensor performance in a number of ways. Firstly, measurement bias is often, but not always, reduced following calibration, as evidenced by a general trend for devices to migrate towards the origin (RMSE = 0 ppb). Secondly, it can help to improve within-manufacturer precision, as evidenced by sensor systems from the same company grouping more closely as the right plot in Fig. 4 shows. The figure also highlights a fundamental challenge with evaluating sensor systems: the measured performance can vary dramatically over time —and space— as the surrounding environmental conditions change. To quantify this, 95% Confidence Intervals (CIs) were estimated for each device using bootstrap simulation and are visualised as a shaded region. For the out-of-the-box data, these regions are noticeably larger than in the calibrated results for most manufacturers, suggesting that colocation calibration has helped to tailor the response of each device to the specific site conditions. This observation suggests that colocation calibration effectively improves each device's response to particular site conditions. This improvement is underscored by the more substantial reduction in the cRMSE component compared to the MBE. The cRMSE, representing the portion of error that persists after bias removal, essentially measures errors attributable to variance within the data space. In the context of out-of-the-box data, this "data space" spans all potential deployment locations used by manufacturers for initial calibration model training (i.e., before shipping the sensors for the QUANT study), thus exhibiting high variability. However, applying site-specific calibration significantly narrows this variability, leveraging local training data to minimise variance.

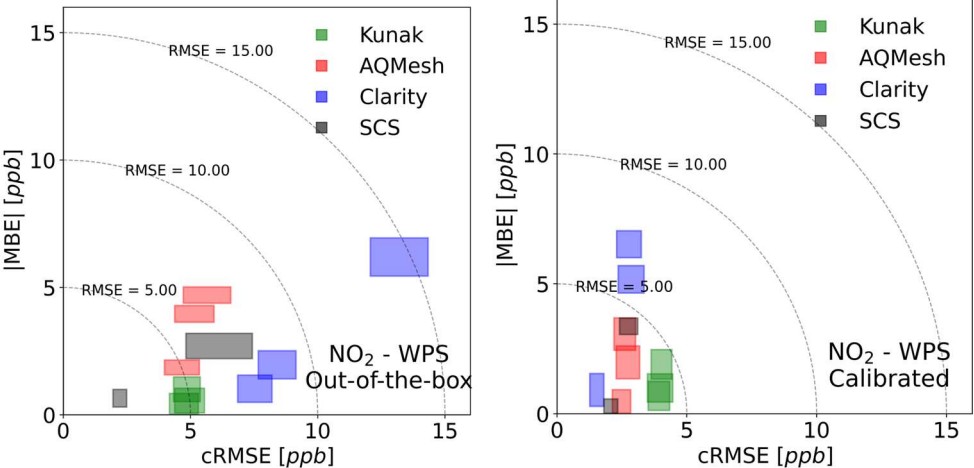

**Figure 4. Effect of colocation calibration on $NO_2$ sensor accuracy. The accuracy is quantified using RMSE, which is decomposed into MBE (y-axis) and cRMSE (x-axis). 95% confidence regions were estimated using bootstrap sampling. The left panel displays results from the period Jun - Jul 2021 ('out-of-the-box' data), while the right-hand panel summarises Aug 2021 when calibrations were applied for all the WPS manufacturers.**

However, it is important to note a limitation of Target Plots: they primarily focus on sensor behaviour around the mean. Therefore, the collective improvement evidenced by Fig. 4 might be only partial. For applications where it is important to understand how calibrations impact lower or higher percentiles, considering other metrics or visual tools would be advisable. An example of this is the absolute and Relative Expanded Uncertainty (REU, defined by the Technical Specification CEN/TS 17660-1:202). Unlike the more commonly used metrics such as $R^2$, RMSE, and MAE, which measure performance of the entire dataset, the REU offers a unique "point by point" evaluation, enabling

its representation in various graphical forms, such as time series or concentration space (for the REU mathematical
derivation, refer to section "S5. Performance Metrics"). The REU approach also incorporates the uncertainty of the
reference method into its assessment, highlighting the intrinsic uncertainty present in all measurements, including
those from reference instruments. This consideration of reference uncertainty is crucial for a holistic understanding of
sensor performance and calibration effectiveness. For a comprehensive discussion on this, refer to Diez et al. (2022).
Fig. 5 illustrates how $NO_2$ calibrations might not only improve collective performance around the mean (as indicated
by the dotted red line in Fig. 5 and previously displayed in the target plot) but across the entire concentration range.

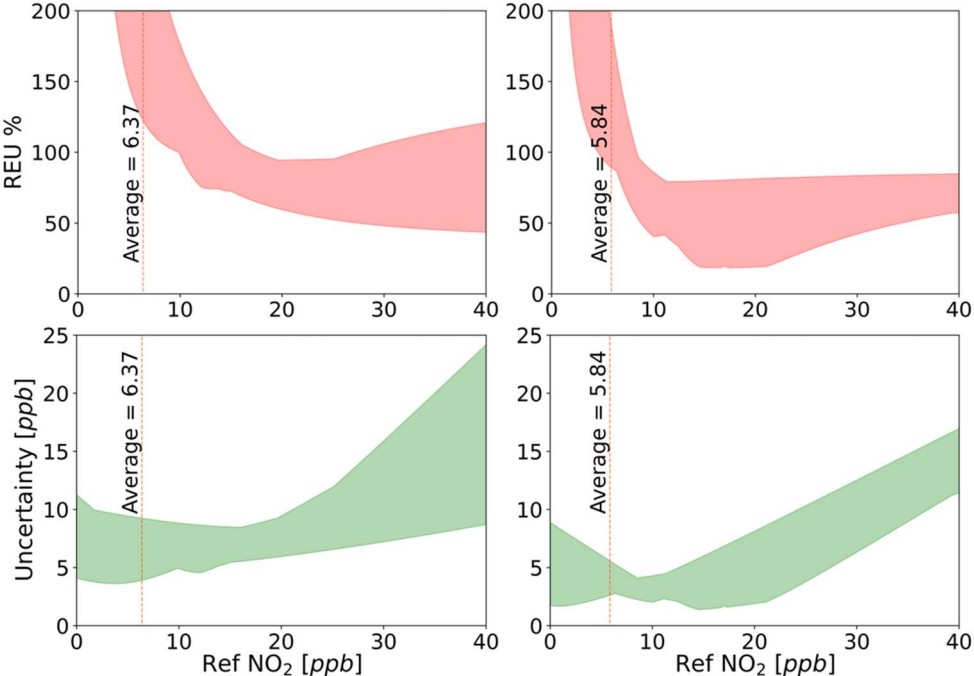


**Figure 5. The top plots display the REU (%) across the concentration range, while the bottom plots depict the Absolute**
**Uncertainty (ppb) —both before (left plots) and after (right plots) calibrating $NO_2$ WPS systems. The shaded areas**
**represent the collective variability evolution (all sensors from all companies) of both metrics. These plots were constructed**
**using the minimum and maximum value of the REU and the Absolute Uncertainty for the entire concentration range.**

However, a note of caution when interpreting results from observational studies such as these is that it is impossible
to ascertain a direct causal relationship between calibration and sensor performance as there are numerous other
confounding factors at play (Diez et al., 2022). Notably these two data products are being assessed over different
periods when many other factors will have changed, for example, the local meteorological conditions as well as
human-made factors such as reduced traffic levels following the COVID-19 lockdown that commenced in March

319 2020.

### 3.3 Reference instrumentation is key

A common assumption when evaluating the performance of sensors is that the metrological characteristics of the
sensor predominantly influence discrepancies detected in co-locations. While this presumption can often be justified
due to both devices' (sensor and the reference method) relative scales of measurement errors, it is not always the case.
Since every measurement is subject to uncertainties, it is crucial to consider those associated with the reference when
deriving the calibration factors of placement.

Fig. 6 (left plots) displays the performance of a $NO_2$ reference instrument (Teledyne T200U) specifically installed for QUANT, located next to the usual instrument at the Manchester supersite (Teledyne T500). Although they use different analytical techniques (chemiluminescence for the T200U and Cavity Attenuated Phase Shift Spectroscopy for the T500), their measurements are highly correlated ($R^2 \sim 0.95$). However, it's possible to identify a proportional bias (slope=0.69), attributed to retaining the initial calibration (conducted in York) without subsequent adjustments, a situation exacerbated by an unnoticed mechanical failure of one of the instrument's components. The REU demonstrates that, under these circumstances, an instrument designated as a reference does not meet the minimum requirements (REU $\leq$ 15% for $NO_2$ reference measurements) set out by the Data Quality Objectives (DQOs) of the EU AQ Directive. Figure S6 shows a unique sensor evaluated against both the T500 and the T200U. The comparison against the T200U yields better results, suggesting that, in a hypothetical scenario where it was the only instrument at the site, this could lead to misleading conclusions. This situation reinforces the idea that instruments should not only be adequately characterised but also undergo rigorous quality assurance and data quality control programs, as well as receive appropriate maintenance (Pinder et al., 2019). All of this must be performed before and during the use of any instrument.

For PM monitoring, the current EU reference method is the gravimetric technique (CEN EN 12341, 2023), which is a non-continuous monitoring method that requires weighing the sampled filters and off-line processing of the results. Techniques that have proven to be equivalent to the reference method (called "equivalent to reference" in the EU AQ Directive) are very often used in practice. In the UK context, the Beta Attenuated Monitor (BAM) and FIDAS (optical aerosol spectrometer) are equivalent-to-reference methods commonly used as part of the Urban AURN Network (Allan et al., 2022). To illustrate these differences in practice, Fig. 6 compares these two equivalent-to-reference $PM_{2.5}$ measurements obtained with a BAM (AURN York site, located on a busy avenue), and a FIDAS unit specifically installed for QUANT. During this specific period, they show a strong linear association ($R^2 = 0.87$). Although the bias is not extremely pronounced (slope=0.80), the FIDAS measurements are, on average, systematically lower compared to BAM.

In the hypothetical case that the BAM were to be considered the reference method (arbitrarily chosen for this example as it is the current instrument at the AURN York site) when assessing the FIDAS under these test conditions, it would only meet the criterion stipulated by the EU DQOs for indicative measurements (REU $\leq$ 25% for $PM_{2.5}$), but not for fixed (i.e., reference) measurements (REU $\leq$ 50% for $PM_{2.5}$). This example is primarily intended to illustrate the magnitude of differences between both methods for this particular application, and by no means does this observation imply that the FIDAS measurements are inherently problematic.

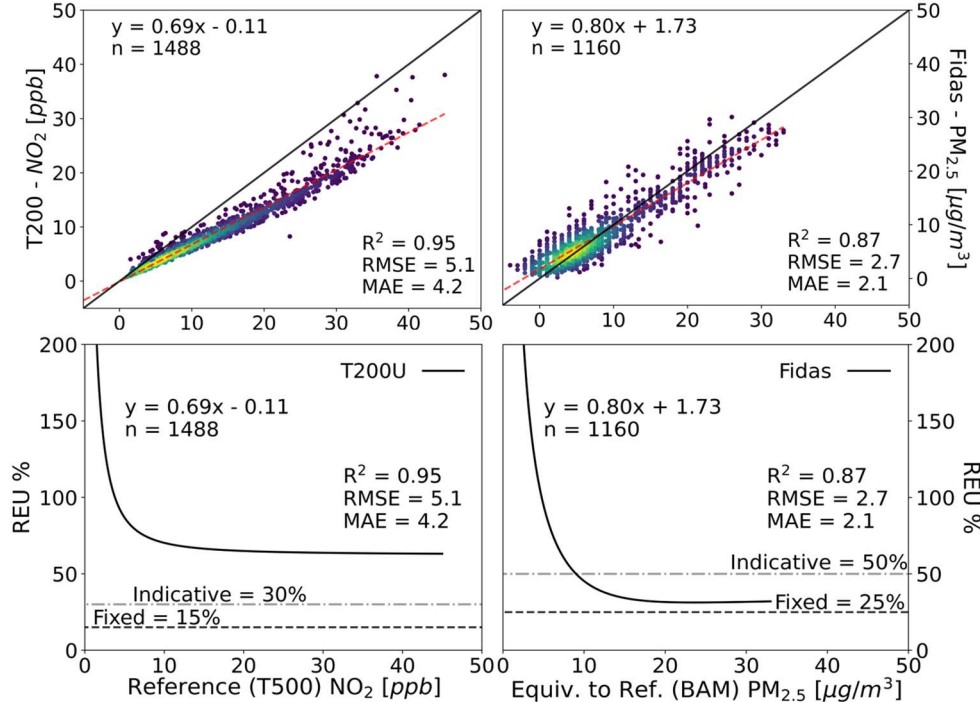

**Figure 6. The left plots depict the comparison between the Teledyne T200U (chemiluminescence analyzer) and the reference method (Teledyne T500 CAPS analyzer) at the Manchester supersite. The plots to the right illustrate PM$_{2.5}$ measurements in York, taken with a FIDAS instrument (optical aerosol spectrometer) and a BAM 1020 (beta attenuation monitor), both equivalent-to-reference methods. While the top plots show the regression (including some typical single-value metrics), those on the bottom present the REU alongside the DQOs defined by the EU AQ Directive.**

Although these two instruments (BAM and Fidas) show a greater concordance between themselves than with sensors (for the comparison of two sensor systems against the BAM and the Fidas, refer to Fig. S74), the choice of the measurement method can have a considerable impact on evaluations of this type. This underscores the importance of adequately characterising the uncertainties of the reference monitor when evaluating sensors.

### 3.4 Inter-location performance

An extreme example of sensor performance varying due to environmental conditions is when sensors are moved between locations, as their apparent performance may vary drastically. Fig. 7 displays the REU and regression plots for four of the same PM$_{2.5}$ sensor system in two periods: April-June 2022 when the devices were working across the 3 sites (York, Manchester and London), and August-October 2022 when they were all reunited in Manchester. The RMSE remains reasonably consistent (range 2.27 to 3.47 ppb) between the devices across the periods and locations. However, for the device that moved from York to Manchester, a change in slope from 0.69 to 0.86 was observed. Because this device's slope is consistent with the other units while running in Manchester, this is likely due to the different sensor responses in the specific environments. The precise cause of this change is not immediately evident and will be the focus of a follow-up study, but could be due to changes in local conditions (e.g., weather, emissions, etc.) impacting sensor calibration and/or differences in actual PM$_{2.5}$ sources and particle characteristics at the sites (Raheja et al., 2022).

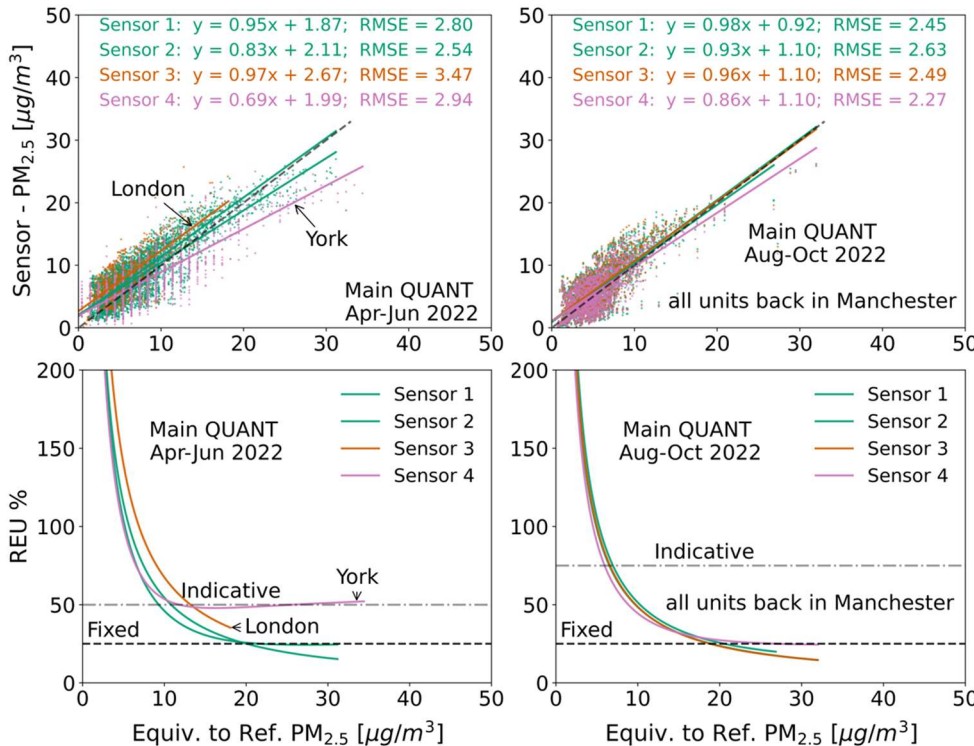

378

**Figure 7. Regression (top) and REU (bottom) plots showing data from four PM$_{2.5}$ sensors (same manufacturer) over 2 time periods: Apr-Jun 2022 and Aug-Oct 2022. The four devices were in separate locations in the first period, but all deployed in Manchester in the second. The horizontal dashed lines represent a reference for the PM$_{2.5}$ DQOs as defined by the EU AQ Directive (for "fixed" PM$_{2.5}$ measurements, REU < 25%; for "indicative" PM$_{2.5}$ measurements, REU < 50%). Readers are encouraged to consult the specified standard for further details.**

A second example of inter-location performance is presented in Fig. 8, showing NO$_2$ data from two sensor systems (from two different manufacturers, identified as Systems A and B) before (left plots) and after (right plots) they were moved from Manchester to London in March 2020. Both sensors saw a reduction in agreement with the reference instrument at the London site compared to Manchester, despite both these sites being classified as urban-background with reference instrument performance regularly audited by the UK National Physical Laboratory.

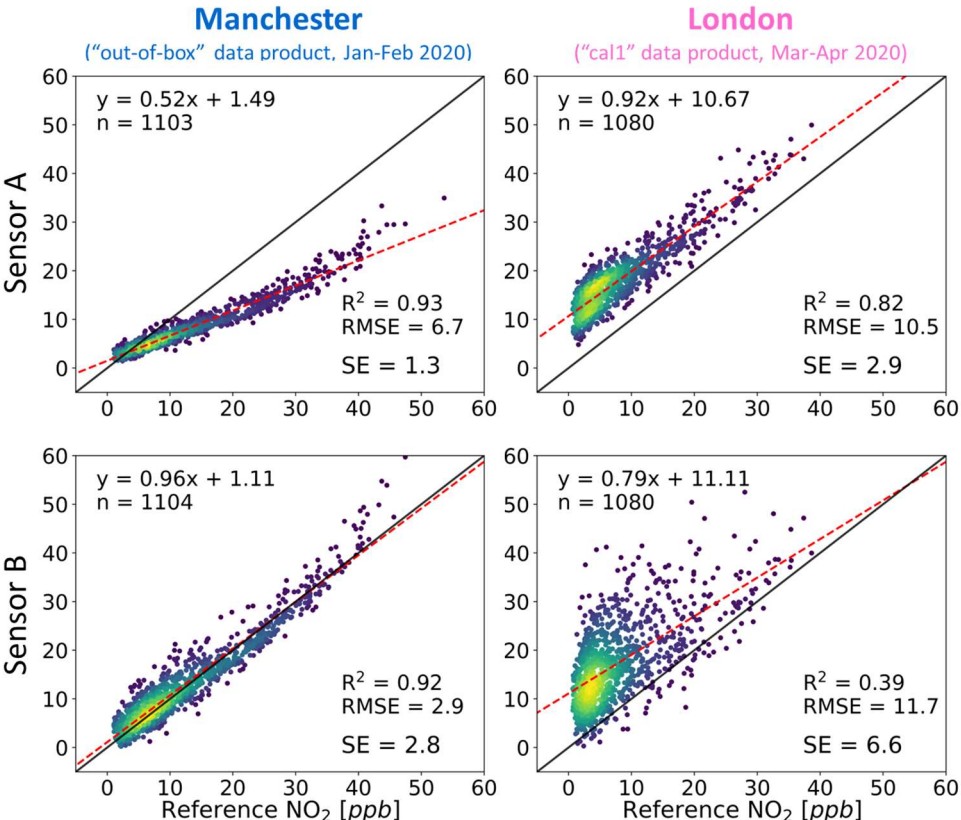

389

**Figure 8. Comparative analysis of NO₂ measurements from two systems (A and B), across two urban settings. The left plots display Manchester "out-of-box" data product (January to February 2020), while the right plots show London "cal1" data product (April to May 2020). This "cal1" label does not indicate corrections specific to London's conditions but denotes a data product from a specific period (as detailed in Figures S2 and S3). The colour gradient represents the density of data points, with darker shades indicating lower densities and brighter shades signifying higher densities.**

The primary distinction between both systems' behaviour lies in the fact that the sensor located in the top row (Sensor A), even after being relocated to London, maintains a linear response (albeit slightly more degraded than that observed in Manchester, as indicated by the $R^2$ and RMSE). In contrast, Sensor B's response becomes significantly noisier upon relocation to London, as highlighted by the Standard Error (SE) —which represents the remaining error after applying a perfect bias correction. Despite both systems utilising identical sensing elements, the variance in residuals between them may stem from the distinct calibration approaches applied by the respective companies.

For cases resembling Sensor A, users might find it beneficial to implement simple linear correction methods (e.g., using reference instruments if available) or explore other strategies for zero and span correction. A practical and cost-effective approach, for example, is using diffusion tubes for NO₂ measurements, as discussed in Section 3.6. Conversely, in scenarios characterised by high variance in residuals, such as those observed with Sensor B, a-posteriori attempts to apply a simple linear correction are unlikely to result in significant improvement. While more sophisticated corrections are theoretically feasible, their effectiveness is limited by the end-user's domain knowledge and the availability of additional complex data sources. Furthermore, it is important to consider that excessive post-processing may lead to overfitting —a situation where a model excessively conforms to specific patterns in the training data, resulting in poor performance on new, unseen data (Aula et al., 2022).

**3.5 Long-term stability**

The long-term stability of sensor response is also an important facet of its performance, especially for certain use cases such as multi-year network deployments. There can be multiple causes of long-term changes to sensor response, for example, particles settling inside the sampling chamber in optical-based sensors(e.g. Hofman et al. (2022)), or the gradually changing composition of electrochemical cells (e.g. Williams (2020)). How these changes manifest themselves in the data must be identified if ways to account for them are to be implemented.

Fig. 9 shows the temporal nature of the $O_3$ and $NO_2$ errors (MBE, cRMSE and RMSE) from a sensor system between February 2020 and October 2022. The $O_3$ shows (Fig. 9a) a gradual increase in the overall measurement error, largely due to an increase in the MBE. It also shows a distinct seasonality MBE, increasing by a factor of 3-4 between March and July compared to the August-February period. The cRMSE component shows fluctuations during the study but only has a small increasing trend. The $NO_2$ system (Fig. 9b) demonstrates a consistently increasing overall error, with a less pronounced seasonal influence. The bias contributes greatly to the total error (see Section 3.6 for $NO_2$ sensor correction, Fig. 9c).

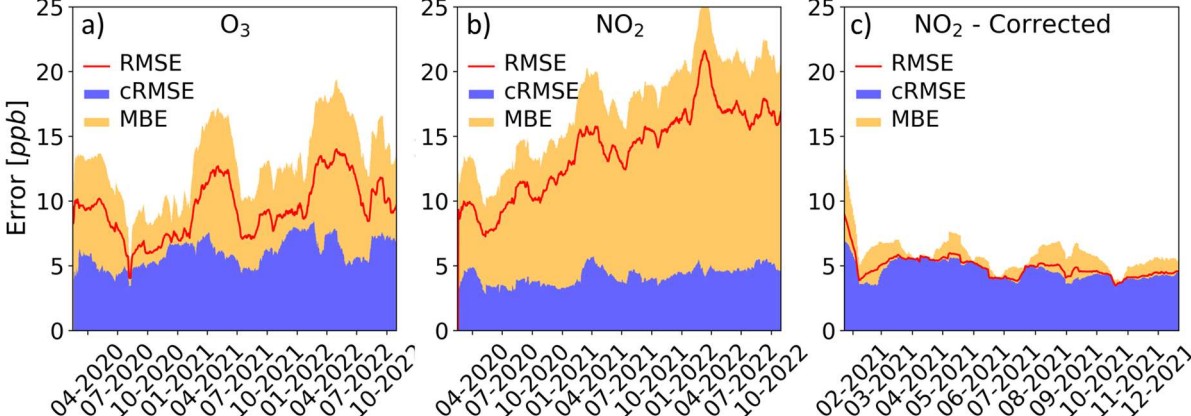

**Figure 9. Seasonal variation of error (as RMSE, red line) of one of the systems belonging to the Main QUANT, decomposed into cRMSE (in blue) and MBE (in yellow) estimated based on a 40-day (aligning with the sample size recommendation by the CEN/TS 17660-1:2021 standard for on-field tests) moving window approach with a 1-day slide (i.e., advancing the calculation 1 day at a time). Panel a) is for $O_3$ measurements, and panel b) is for $NO_2$ (April 2020-Oct 2022). Panel c) is also for $NO_2$, this time showing the effect of a linear correction using diffusion tubes (see next section for more details).**

## 3.6 Informing end-use applications

Ultimately, for any air pollution monitoring application, the requirements of the task should dictate the measurement technology options available. For example, if the requirement for a particular measurement is to assess legal compliance, then lower measurement uncertainty must be a key consideration as the reported values need to be compared to a limit value. In contrast, if an application aimed to look at long-term trends in pollutants, then absolute accuracy may not be as important as the long-term stability of sensor response. To realise the potential of air pollution sensor technologies, end users need to align their specific measurement needs with the capabilities of available devices. Achieving this necessitates access to unbiased performance data, such as long-term stability and accuracy across varying conditions, ideally in an easy-to-access and interpret manner.

Understanding the uncertainty associated with a instrument is essential for recognizing its capabilities and limitations. Accurate instruments are crucial, especially in areas like public health decision-making, where inaccurate data can have profound implications (Molina Rueda et al., 2023). Furthermore, instruments that operate autonomously ensure consistent, uninterrupted data collection, making them more efficient and cost-effective in terms of maintenance and calibration. Figure 10 illustrates the collective behaviour of $NO_2$ sensors from each of the four companies with more than two working systems, showcasing their REU (y-axis) versus Data Coverage (DC, x-axis). Both parameters were calculated for each sensor system using a 40-day moving window approach and then aggregated by brand, ensuring a comprehensive analysis. This methodology leverages overlapping data from multiple sensors to provide a robust representation of company-wide sensor performance and aims to prevent biassed interpretations. Both REU and DC are key criteria within the EU scheme (EU 2008/50/EC) for evaluating the performance of measurement methods, and are complemented by the CEN/TS 17660-1:2021 specifically for sensors. The latter document defines three different sensor system tiers. Class 1 $NO_2$ sensors, bounded by the green rectangle (REU < 25% and DC > 90%), offer higher accuracy than Class 2 sensors (REU < 75% and DC > 50%), delimited by the red rectangle (Class 3 sensors have no set requirements). Presenting the REU and DC like in Fig. 10 helps users anticipate the performance of sensor systems —under the assumption that all sensors from the same brand will behave similarly in equivalent environmental conditions— providing more insight into selecting the appropriate instrument for a given project or study.

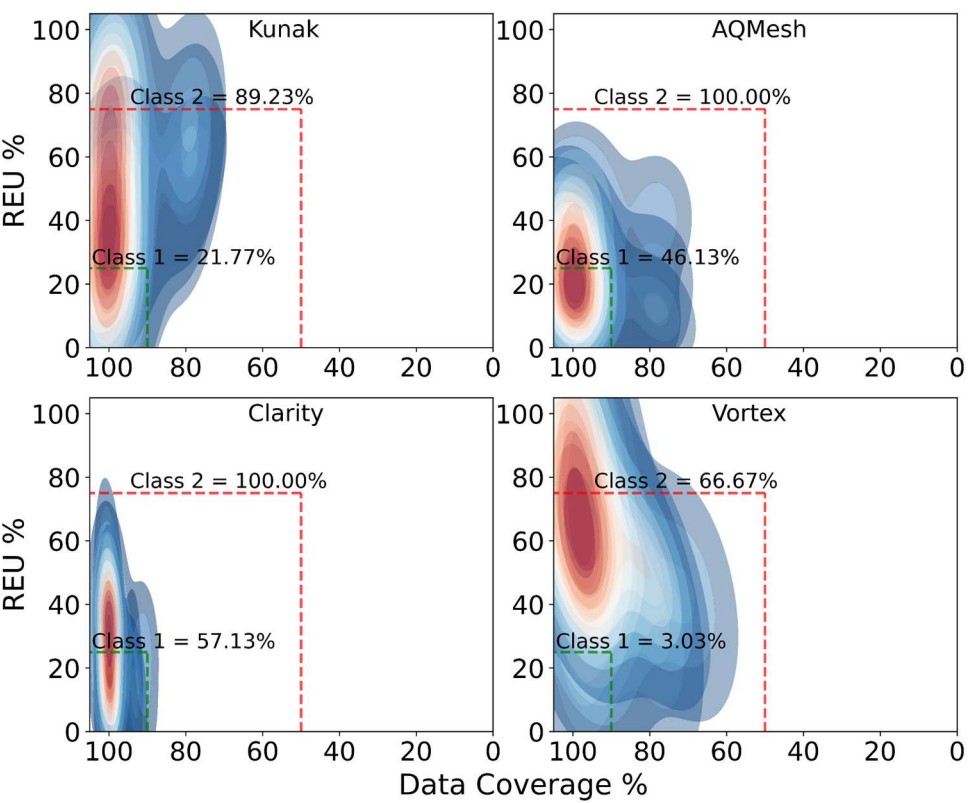

**Figure 10. REU vs. DC for 4 sensor system companies measuring NO₂, with more than two units working simultaneously during the WPS (period Nov 2021-Oct 2022, after all companies provided at least one calibrated product). Each heat map plot (cooler colours for lower densities and warmer colours for higher densities) aggregates the REU and DC from sensors of the same brand working concurrently. The calculation of these two parameters employ a 40-day (aligning with the sample size recommendation by the CEN/TS 17660-1:2021 standard for on-field tests) moving window approach with a 1-day slide (i.e., advancing the calculation 1 day at a time). The green dashed rectangle limits the Data Quality Objectives (DQOs) for Class 1 sensors, and the red dashed rectangle outlines the DQOs for Class 2 sensors.**

Depending on the nature of the sensor data uncertainty, methods can be implemented to improve certain aspects of the data quality for a particular application. One such example is the use of distributed networks to estimate sensor measurement errors, such as that described by (J. Kim et al., 2018). Depending on the application and available options, users can access alternative methods to reduce bias, thus enhancing the accuracy of sensor systems and networks. For example, "Indicative methods", as defined by the EU AQ Directive, such as diffusion tubes (e.g., NOx, SO$_2$, VOCs, etc.), can be an option. Specifically, our study leverages diffusion tube data for NO$_2$, illustrating one effective approach to bias correction using supporting observations, as exemplified in Fig. 9b. These measurements are widely used to monitor NO$_2$ concentrations in UK urban environments, due to their lower cost (~£5 per tube) and ease of deployment, but only provide average concentrations over periods of weeks to months (Butterfield et al., 2021). During QUANT, NO$_2$ diffusion tubes were deployed at the 3 colocation sites (see Section S7 at the Supp. for more details). Combining these measurements offers the possibility of quantifying the average sensor bias, thus reducing the error on the sensor measurement whilst maintaining the benefits of its high time-resolution observations. It is important to note that while bias correction has been applied to the sensor data, the NO$_2$ diffusion tube concentrations used for comparison purposes must also be adjusted (e.g. following Defra (2022)). Fig. 9c shows the accuracy of the same NO$_2$ sensor data shown in Fig. 9b but applies a monthly offset calculated as the difference between its monthly average measurement and that from the diffusion tube (see Figure S8). This shows a dramatic reduction in overall error largely driven by its bias correction. What remains largely resulting from the cRMSE, i.e. the error variance that might arise from limitations from the sensing technology itself and/or the conversion algorithms used to transform the raw signals into the concentration output. To validate the efficacy and reliability of this bias correction method, further long-term studies are warranted.

The development and communication of methods that improve sensor data quality, ideally in accessible case studies, would likely increase the successful application of sensor devices for local air quality management. There is also a need for similar case studies showcasing the successful application of sensor devices for particular monitoring tasks. An example of this from the QUANT dataset is the use of sensor devices to successfully identify change points in a pollutant's concentration profile. These are points in time where the parameters governing the data generation process are identified to change, commonly the mean or variance, and can arise from human-made or natural phenomena (Aminikhanghahi and Cook, 2017). Determining when a specific pollutant has changed its temporal nature is a challenging task as there are a large number of confounding factors that influence atmospheric concentrations, including but not limited to seasonal factors, environmental conditions (both natural and arising from human behaviour), and meteorological factors. This challenge has lead to several "deweathering" techniques being proposed in the literature (Carslaw et al., 2007; Grange and Carslaw, 2019; Ropkins et al., 2022). While change point detection is highlighted here as a promising application of sensor data, it represents just one of many potential methodologies that could be explored with the QUANT dataset.

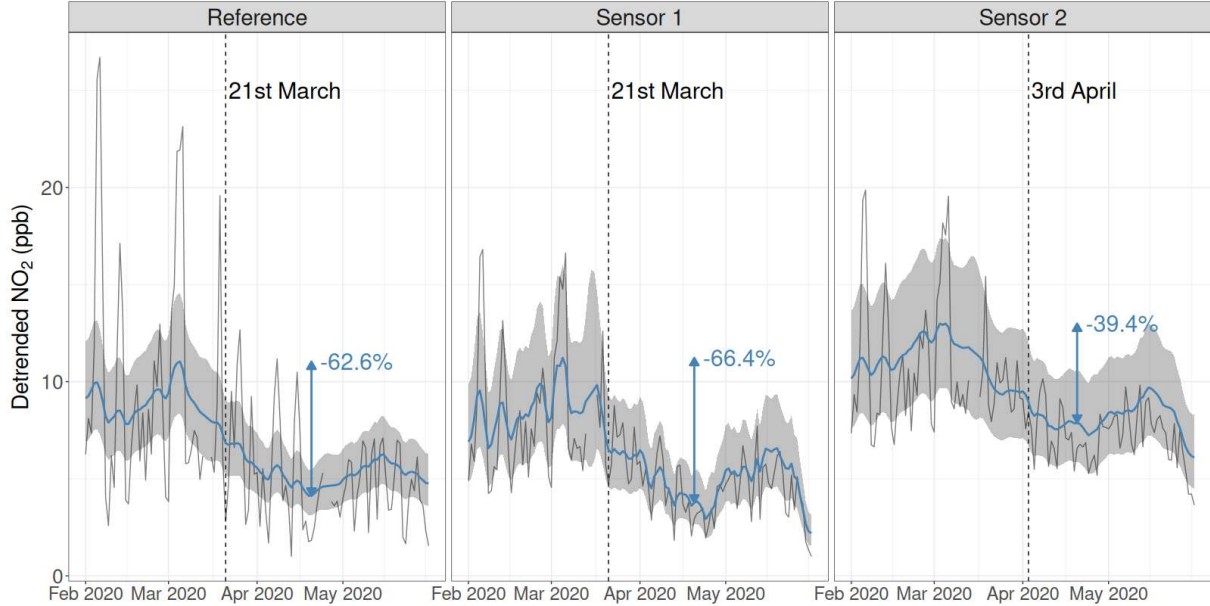

496

**Figure 11. NO₂ measurements (black solid line) and detrended estimates (blue solid line with 95% confidence interval in the shaded grey region) from the reference instrument (left panel) and 2 sensor systems (middle and right panels) from Manchester in 2020. Vertical dashed lines and their corresponding dates indicate identified change points, which correspond to the introduction of the first national lockdown due to COVID-19 on the 23rd of March 2020. The percentage in blue represents the relative peak-trough decrease from 5th March to 20th April.**

A state-space based deweathering model was applied to NO₂ concentrations measured from the sensor systems that had remained in Manchester throughout 2020 to remove these confounding factors, with the overarching objective to identify whether the well-documented reduction in ambient NO₂ concentrations due to changes in travel patterns associated with COVID-19 restrictions could be observed in the low-cost sensor systems. To provide a quantifiable measure of whether a meaningful reduction had occurred, the Bayesian online change-point detection (Adams & MacKay, 2007) was applied. Of the 8 devices that measured NO₂, clear change points corresponding to the introduction of a lockdown were identified in 2 (Fig.11), demonstrating the potential of these devices to identify long-term trends with appropriate processing, even with only 3 months of training data.

## 4. Conclusions

Lower-cost air pollution sensor technologies have significant potential to improve our understanding and ability to manage air pollution issues. Large-scale uptake in the use of these devices for air quality management has, however, been primarily limited by concerns over data quality and a general lack of a realistic characterisation of the measurement uncertainties making it difficult to design end uses that make the most of the data information content. Advances are occurring rapidly, in both the measurement technology and particularly in the data post-processing and calibration. A challenge with the use of sensor-based devices is that many of the end-use communities do not have access to extensive reference-grade air pollution measurement capability (Lewis & Edwards, 2016), or in many cases, expertise in making atmospheric measurements or the technical ability for data post-processing. For this reason, reliable information on expected sensor performance needs to be available to aid effective end-use applications. Large-scale independent assessments of air sensor technologies are non-trivial and costly, however, making it difficult for end users to find relevant performance information on current sensor technologies. The QUANT assessment is a multi-

year study across multiple locations, that aims to provide relevant information on the strengths and weaknesses of commercial air pollution sensors in UK urban environments.

The QUANT sensor systems were installed at two highly instrumented urban background measurement sites, in Manchester and London, and one roadside monitoring station in York. The study design ensured that multiple devices were collocated to assess inter-device precision, and devices were also moved between locations and able to test additional calibration data products to assess and enable developments in sensor performance under realistic end-use scenarios. A wider participation component of the Main QUANT assessment was also run at the Manchester site to expand the market representation of devices included in the study, and also to assess recent developments in the field.

A high-level analysis of the dataset has highlighted multiple facets of air pollution sensor performance that will help inform their future usage. Inter-device precision has been shown to vary, both between different devices of the same brand and model and over different periods of time, with the most accurate devices generally showing the highest levels of inter-device precision. The accuracy of the reported data for a particular device can be impacted by a variety of factors, from the calibrations applied to its location or seasonality. This has important implications for the way sensor-based technologies are deployed and supports the case made by others (Bittner et al., 2022; Farquhar et al., 2021; Crilley et al., 2018; Williams, 2020; Bi et al., 2020) that practical methods to monitor sensor bias will be crucial in uses where data accuracy is paramount. Ultimately, this work shows that sensor performance can be highly variable between different devices and end-users need to be provided with impartial performance data on characteristics such as accuracy, inter-device precision, long-term drift and calibration transferability in order to decide on the right measurement tool for their specific application.

In addition to these findings, this overview lays the groundwork for more detailed research to be presented in future publications. Subsequent analyses will focus on providing a more nuanced understanding of the uncertainty in air pollution sensor measurements, thus equipping end-users with better insights into the capability of sensor data. Future studies will delve into specific aspects of air pollution sensor performance: 1) a comprehensive performance evaluation of $PM_{2.5}$ data, assessing their accuracy and reliability under different environmental conditions; 2) an in-depth analysis of $NO_2$ measurements, examining their sensitivity and response in various urban environments; and 3) a detailed investigation into the detection limits of these sensor technologies, targeting their optimised application in low concentration scenarios. These focused studies are basic steps needed to further advance our understanding of sensors' capabilities and limitations, ensuring informed and effective application in air quality monitoring.

**Supplementary**

The supplement related to this article is available online at:

**Data availability**

The QUANT dataset, accessible at the Centre for Environmental Data Analysis (CEDA) (Lacy et al., 2023; https://catalogue.ceda.ac.uk/uuid/ae1df3ef736f4248927984b7aa079d2e), is the most extensive collection to date assessing air pollution sensors' performance in UK urban settings. It encompasses gas and PM sensor data recorded in the native reporting frequency of each device. The reference data from the three monitoring sites can be found at:

- MAQS: https://data.ceda.ac.uk/badc/osca/data/manchester;
- LAQS: https://www.londonair.org.uk/london/asp/datadownload.asp;

●   YoFi: https://uk-air.defra.gov.uk/data/data_selector.
A comprehensive data descriptor manuscript, detailing the QUANT dataset's collection methods, processing
protocols, accessibility features, and overall structure—including variables, data reporting frequencies, and QA/QC
practices—has been submitted for publication. At the time of this writing, the manuscript is still under review.
A GitHub repository at https://github.com/wacl-york/quant-air-pollution-measurement-errors provides access to
Python and R scripts designed for generating diagnostic visuals and metrics related to the QUANT study, along with
sample analyses using the QUANT dataset.
**Author contributions**
The initial draft of the manuscript was created by SD, PME, and SL. The research was conceptualised, designed, and
conducted by PME and SD. Methodological framework and conceptualization were developed by SD, PME, and SL.
Data analysis was primarily conducted by SD and SL. The software tools for data visualisation and analysis were
developed by SD and enhanced by SL. MF, MP and NM supplied the reference data critical for the study. TB, HC,
DH, SG, NAM and JU made substantive revisions to the manuscript, enriching the final submission.
**Competing interests**
The authors declare that they have no conflict of interest.
**Acknowledgements**
This work was funded as part of the UKRI Strategic Priorities Fund Clean Air program (NERC NE/T00195X/1), with
support from Defra. We would also like to thank the OSCA team (Integrated Research Observation System for Clean
Air, NERC NE/T001984/1, NE/T001917/1) at the MAQS, for their assistance in data collection for the regulatory-
grade instruments. The authors wish to acknowledge Dr. Katie Read and the Atmospheric Measurement and
Observation Facility (AMOF), a Natural Environment Research Council (UKRI-NERC) funded facility, for providing
the Teledyne 200U used in this study and for their expertise on its deployment. Special thanks are due to Dr David
Green (Imperial College London) for granting access and sharing the data from LAQS (NERC NE/T001909/1).
Special thanks to Chris Anthony, Killian Murphy, Steve Andrews and Jenny Hudson-Bell from WACL for the help
and support to the project. Our acknowledgment would be incomplete without mentioning Stuart Murray and Chris
Rhodes from the Department of Chemistry Workshop for their technical assistance and advice. Further, we
acknowledge Andrew Gillah, Jordan Walters, Liz Bates and Michael Golightly from the City of York Council, who
were instrumental in facilitating site access and regularly checking on instrument status. We acknowledge the use of
ChatGPT to improve the writing style of this article.

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
