# Peer review of "Long-term Evaluation of Commercial Air Quality Sensors: An"

_Atmospheric Measurement Techniques, 2023_

## Referee Comment (RC1)

**Long-term Intercomparison of Commercial Air Quality Sensors: An Overview from the QUANT Study**

Sebastian Diez, Stuart Lacy, Hugh Coe, Josefina Urquiza, Max Priestman, Michael Flynn, Nicholas Marsden, Nicholas A. Martin, Stefan Gillott, Thomas Bannan, Pete Edwards

This study is a comprehensive, long-term study of a wide range of air sensors currently available in the market. Because of the collocation in time and space during the comparison, environmental variables are lessened thereby focusing the comparison on the performance of the air sensors relative to one another and in comparison, to a "reference" monitor. This adds great value to the study. The sensors were deployed and collocated in a real-world environment that they will most likely be used, so the conditions at which the sensors are compared were not biased because of the environment (i.e., compared to if it were collocated in a "pristine" environment, or under laboratory conditions). This study includes both gas and PM sensors which adds to the novelty of the study.

This paper is recommended for publication in AMT with revisions as outlined in this discussion.

**GENERAL COMMENTS**

-   The paper can benefit from tightening up language and being more succinct and concise in its statements.
-   A glossary of terminologies for commonly used yet widely misused or confused terms in the field (e.g., "sensor" vs "sensor systems" vs "sensing unit", "manufacturer" vs "company", "model"/"unit"/"type"/"device") may be useful to the reader, also to help guide the authors in using consistent terminology all throughout the paper.
-   A major issue is the description and reliance on usual metrics like $R^2$ in comparing two instrumental methods. With a goal of prediction and calibration in mind, $R^2$ is an appropriate statistical metric; however, in plainly comparing the correlation (specifically: the concordance or agreement between two measurements), the authors are recommended to use more appropriate statistical metrics that measure ***concordance*** such as the concordance correlation coefficient. The paper can also greatly benefit a wider audience if the authors expanded on the statistical discussion and provide a separate discussion of the statistical metrics used, thus serving as a technical guidance that outlines metrics that can be used in such an intercomparison or calibration exercise.

**SPECIFIC COMMENTS**

-   For the title: A more specific term than "Evaluation" can be used. Suggestions: "Intercomparison"; "Precision Analysis"
-   The abstract is missing some key findings and results, e.g. how many air sensors and reference sensors were quantified, statistical metrics used to quantify the performance of the sensors, etc. For example, Lines 89-90 can be added to the abstract.
-   A separate section on the methodology summarizing performance metrics used, and explaining each under a subheader, e.g., "Bias" as a subheader and explaining R2, RMSE, etc. under this

heading would be useful to the reader and also makes this a good reference paper for intercomparison studies.

- Section 3.3 explores reference instrumentation. Authors need to make what is meant by "reference", e.g. a reference method designated by an authority (EU, US EPA, etc.) or a self-defined or agreed-upon reference method.
- In one section, the manufacturers / models of air sensors were referred to; however, in Figure 10, it was anonymized. What was the rationale? Can it be consistently anonymized or named? And if not, explain why and make sure that the transition is clearly explained within each section.
- It is useful from a consumer perspective to mention which devices are available readily as-is (without add-ons) and/or which ones require customization from the manufacturer's end. This can possibly be added to the summary in Table 1 and/or Supplementary S1, with a short reference (a sentence or two) in the main text.
- Might be useful to add in the conclusions / recommendations section for future researchers: quantify inter-location variability.
- Might be useful to explain and emphasize (including in the abstract) why correction with satellite data was not explored in this study.
- Can employ the terms "inter-device" and "inter-location" for succinctness of ideas.

**Line by line comments**

- Line 45: "cross-sensitivity" seems to be a term usually used in the medical context. It might be beneficial to define this term in this context, and differentiate it from "interference". Levy Zamora (2022) used "cross-sensitivity" in the title of their article and Bittner (2022) defined it, so it might be helpful if these two comes as the first articles cited in this instance)
- Lines 48-49: citations for temperature and humidity might be combined since they are usually explained in cited references in combination.
- Lines 63-68: This paragraph could benefit from differentiating "calibration" from "correction" and how these two terms are sometimes interchangeably used (albeit incorrectly). A reference to an article that explains this difference will also be helpful. The Liang (2022) paper cited explains some of these nuances, including mathematical equations for calibrations, but does not fully differentiate "correction" from "calibration".
- Lines 67-68: True for gases. Mention examples of acceptable calibration method(s) for PM?
- Line 73: See also Raheja, et al (2023) https://doi.org/10.1021/acs.est.2c09264
- Lines 81-83: Another reason is that there are a lot of sensors/sensor systems with different configurations commercially available, and also individual sensing units are sold and can be "DIY"-ed—the market is diluted with many options and many different iterations of the same underlying technology with marginal differences.
- Line 94: Clarify or add examples of "data products" e.g., APIs, mobile apps, etc.
- Lines 105-106 and 116: Useful to add a subsection that describes the UK urban environment including seasonality, sources of pollution (transportation? Household commercial products use?) in the three locations (London, Manchester and York)
- Line 106: "replicates" or "units" are more appropriate terms than "duplicates" if you are talking of the units of the same model
- Line 109: define what is meant by "near real-time" in this context.
- Line 113: Were the units tested together before deploying separately? Clarify.

- Line 121: A sentence or two succinctly describing the sites will also be useful in this line. Then you can refer to the Supplementary.
- Lines 122-126: Consider moving up before Lines 113-121.
- Line 125: "inter-device consistency" may also be rewritten as "precision".
- Lines 134-136: "vendors were invited to contribute multiple sensor devices throughout the WPS study". How does a "sensor device" differ from a "sensor" or "sensor system"? Does this mean that manufacturers can contribute different sensor models? Also, does vendor = company = manufacturer? Note consistent terminology all throughout the manuscript (might be useful to have a glossary or footnote, like that for "sensor" and "sensor system" on page 2.
- Lines 139-141: Table 1. Does AQMesh AQM, Kunak AP, and SCS Prax have *all* of the sensors listed (from NO to PM10) in one unit? This table might benefit from a clarification (can be added to the Table caption). Also, as mentioned in a previous comment, add in the description if these are consumer-ready (eg already sold in the market as that unit), or customizations available from the manufacturer.
- Lines 148-149: I understand that PurpleAir does not have a mobile data connection, only WiFi, but WiFi was not good in the location so you opted to download the data from the device memory instead. The text can be enhanced by better explaining the issue as described. (i.e., differentiating from WiFi and mobile data connectivity)
- Line 150: and harmonize? In the methodology section, it might be useful to mention that temporal and spatial scales of the sensor systems was important to match, thus aggregation and harmonization was necessary. How was incomplete data treated? Were there imputed data? Might be useful to add it in the supplementary.
- Line 150: by data format, do you mean datetime / time and date?
- Line 158: "calibrated data products": is this referring to API? Measurements? As with my previous comment – clarify what "data products" mean.
- Lines 160-166: What is cal1? cal2? Clearly define / describe these in the text and/or supplementary. This section may benefit from a subsection explaining / describing these.
- Lines 170-174: Is this a caveat / weakness of using these statistical metrics used herein ($R^2$, MAE, etc)? What is the alternative? I suggest concordance (agreement) metrics, such as the Concordance Correlation Coefficient: See Lin, Biometrics (1989): https://doi.org/10.2307/2532051. The reader might also benefit from a separate subsection and/or supplementary section describing the metrics or including a glossary of the metrics used.
- Line 183: "multiple devices of the same type" when you mean "type" do you mean similar underlying principles of measurement? Model? Be consistent in terminology. Also, it might be useful to cite an example of which devices you are considering a same "type", e.g. AQM and Clarity—are these of the same "type" as described?
- Lines 190-196. See also deSouza (2023): An analysis of degradation in low-cost particulate matter sensors https://doi.org/10.1039/D2EA00142J
- Lines 202-204: Good point.
- Line: 217: 75% inclusion criteria is common—but perhaps not for readers not familiar with this data type. Readers might benefit from a citation, explanation in methodology or supplementary. Suggested section to add it in: Section 2.1, lines 150-151.
- Line 225: Clarify: did you mean closer together spatially / physical location? How "close" is close?
- Line 232: could benefit more from a further explanation of the bias-variance tradeoff.

- Section 3.3. and Supplementary S4. Cite the authorities that consider the instruments mentioned as reference. e.g., are they considered "reference" because they are listed in an EU directive or US EPA documentation? If so, cite these. If not, provide a rationale or a citation as to how these instruments were categorized as "reference".
- Lines 241-242: Expound on the significance and advantages of REU as opposed to the other metrics.
- Lines 268-269: clarify / reiterate the said minimum requirements in this text.
- Lines 283-284: $R^2$ can many times be subjective, e.g. how can we say that an $R^2$ of 0.87 means "it does not fully agree" and a slope of 0.80 is considered a ***pronounced*** bias? This is where the definition of concordance and using concordance metrics might be useful. If two measurements are concordant (in agreement), then slope is expected to be unity (=1). Also, a high $R^2$ does not necessarily mean agreement between the two instruments. Also, clarify what is meant by "limiting the linearity". The authors are cautioned against using $R^2$ in quantifying agreement between two instruments that are being compared.
- Lines 288-289: Specify criterion stipulated by EU DQOs
- Line 308: "reasonably consistent" – reasonably is subjective and qualitative. Suggest dropping the word, or provide a percentage of the time that the RMSE is consistent (e.g. provide a qualitative measure).
- Line 312: "local conditions": give or name some examples. Do you mean weather conditions? Traffic? etc.
- Line 334: quickly define (add a phrase) that describes "overfitting"
- Line 336: "linear correction" – "linear regression" might be the appropriate term.
- Line 347: RMSE also showed seasonality.
- Lines 345-351: Add more explanations about the seasonality. Add recommendations.
- Line 355: Clarify what a "1-day slide" means – it can be added in the supplementary or a quick description in the figure caption.
- Lines 363-366: This can be said more succinctly. Also, what sort of information can be provided? Be specific, based on your results so far—what sort of information can you recommend be provided?
- Line 373: Inconsistency in the use of the term "sensor", "system" and "sensor system".
- Lines 377-379: This discussion can benefit from a more detailed explanation of the tiers (Classes) assigned and what was the basis of the assignment to different classes. The figure caption for Figure 10 offers an explanation, which should be repeated and explained in more detail in-text.
- Line 388 onwards can benefit from a separate subheader / subsection.
- Lines 390-391: Specify an example of "simpler methods"
- Line 393: explain more by what instrumental method is $NO_2$ measured/detected from these diffusion tubes. Cite a reference as well.
- Line 411: "change points": does this mean inflection points? Periods of biggest slopes? Peak concentration periods? Consider changing verbiage.
- Lines 414-415: Is it applied in this paper? If so, this line should be described in the methodology and explained further.
- Line 421: Expound what "unsupervised analysis" means in this context. General verbiage related to machine learning may sometimes be unnecessary to use in this text, and can be avoided,

because fundamental/rudimentary statistical metrics (as opposed to complex "black-box" machine learning algorithms) are used.

- Line 422: consistency in terminology. Do the authors mean "sensor system" when they mention "devices"?
- Line 433: "..use of these devices has been primarily limited…" I would disagree, because consumers and many users still use these devices (sensor systems) and they aren't necessarily limited by accuracy concerns, e.g. many users are willing to accept a large margin of error for awareness purposes.
- Line 439: suggested addition: (limitations in) technical ability in post-processing of data
- Lines 460-461: Will this be done by the authors in a future study, or is this a call/recommendation for other researchers?
- Line 466: suggestion for a future study: explore different VOC-$NO_x$ regimes (see Wennberg, ES&T Air: https://doi.org/10.1021/acsestair.3c00055)

**TECHNICAL CORRECTIONS**

Grammatical, Typographical, Figure and Formatting comments

**Throughout the text**

- Note the usage of "data" as a plural noun, e.g. "data were" rather than "data was"
- "Manufacturer" rather than "Company" might be a more descriptive noun for the intended usage.
- "co-location" vs "collocation"? Stay consistent.
- Many links in the "References" section of the supplementary point to a Zotero page that is meant for Google docs, thus rendering the links inaccessible

**Figure comments**

In general, labeling the figure panels with letters (e.g. (a), (b), (c), (d)) allows for easier and clearer reference in text and in figure captions. (e.g. Line 327 mentions the "top row" in Figure 8)

- Figure 1. Good visual—a nice representation of the timeline of events.
- Figure 7. Which sensors are being compared here? Why the anonymity compared to the other section(s)? Also, the readers may benefit from a colorblind-friendly and more contrasting color palette. "Class 1" and "Class 2" sensors are not actually described until page 15 (line 377 onwards) – it might be useful to refer to this section (i.e. Section 3.6) in the figure caption or the accompanying text (paragraphs) that describes this figure, and mention that it will be thoroughly explained in that section.
- Figures 8 and 10. Explain the colorations, e.g. is it meant to be a heat map? What do the specific colors mean? Figure 10 may also benefit from higher contrasting – difficult to see the contrast especially in the lower left panel, and when the plots are printed. Dashing is also difficult to see—might benefit from greater color contrast.

**Line by Line**

- Line 80: Suggestion: "academia" or "academic research" instead of "academic arena".
- Line 104: Suggestion: reword "transparent". Suggested synonyms: open, comprehensive (this changes the meaning a bit)
- Line 118: Typo: "influenced"
- Line 155: Suggestion: reword "ratified" to "validated"
- Line 160: is "time-line" the correct term? Perhaps "comparison" or "matrix" would be more apt for Figure S1; Figure S2 is a scatter plot or a bivariate plot.
- Line 162: change "to use this data" to "to use **these** data"
- Line 206: "Mean Bias Error" rather than "Mean Error Bias"
- Line 209: enclose "out-of-box" in quotation marks; typically "out-of-the-box"
  Line 231: semi-colon after "MBE", comma after "machine learning"
- Line 257: actual "metrological" as in measurements and units, or "meteorological" as in RH and Temp?
- Line 271: "…hypothetical scenario where it…" does "it" refer to T200U? T500?
- Line 275: "All of this" to "all of **these**"
- Line 276: Add comma after "monitoring"
- Line 278: "equivalent-to-reference" – consistency in hyphenation
- Line 282: "obtained with a BAM **at the** AURN York site, located on a busy avenue" – delete parentheses
- Line 289: Omit "of course"
- Line 299: capitalize "FIDAS"
- Lines 300-301: the choice of the reference measurement
- Line 309: Paraphrase "saw its slope change". Suggested: …"a slope change from 0.69 to 0.86 was observed…"
- Line 310: change "when" to "while"
- Line 321: "despite" might not be the correct conjunction here.
- Line 331: change "akin to **this** latter" to "akin to **the** latter"
- Line 268: Redundant. Change "with a measurement instrument" to "with an instrument"
- Lines 368-369: Can be paraphrased to be more succinct.
- Line 383: "4-system" rather than "4 systems companies"
- Line 386: add "dashed", i.e., green dashed rectangle
- Line 398-399. "high time-resolution" (note hyphen placement)
- Line 400: subscript on $NO_2$
- Line 400: Is "DEFRA" all capitalized, or is it "Defra" as mentioned in the acknowledgement (Line 489)?
- Line 407: Consider using a different word from "digestible"
- Line 433: change "uptake" to "uptick"
- Line 452: "high level" seems unnecessary.
- Line 455: "accuracy with respect to reference methods"
- Line 471: Lacks the link to supplementary information (online version link is accessible).

---

## Referee Comment (RC3)

**ARTICLE**

**Long-term Evaluation 1 of Commercial Air Quality Sensors: An Overview from the QUANT Study**

Sebastian Diez, Stuart Lacy, Hugh Coe, Josefina Urquiza, Max Priestman, Michael Flynn, Nicholas Marsden, Nicholas A. Martin, Stefan Gillott, Thomas Bannan, Pete Edwards

https://doi.org/10.5194/amt-2023-251

**GENERAL COMMENTS**

This article provides an important contribution to the advancement of studies associated with air quality sensors. It provides a good overview and information about the QUANT study and some important results. Discussions associated with data quality add value in an important way to alert to errors and possible corrections associated with time and space. Shows the importance of using reference sensors in calibrations, detailing correct use and necessary considerations.

I recommend this publication. However, as this is an important study that can be reused or used as a basis for others, I think it is important to go into more detail, especially methodologically, so that it can be continued and used, as the authors suggest at the end.

**SPECIFIC COMMENTS**

**SECTION 2**

**Section 2.1**

As spatial analyzes are carried out, I consider the spatial description of the areas of the article to be important, such as distances and spatial layout. A spatial image would enhance spatial visualization and discussion. This arrangement is important in analyzing spatial differences and environmental conditions that influence the data.

**Section 2.3**

Line 135. The sensors were implemented according to the manufacturer's specifications. Was any standardization found in the logistics or studied at this stage? I think it's important to describe this stage, perhaps in supplementary material. The layout of the

sensors, whether it was completely open or needed some protection, ground height, proximity to the reference, obstacles, necessary infrastructure, etc. These are all factors that influence the data and are still the subject of much discussion when it comes to implementing the sensors.

**Section about treatments, analysis, and metrics**

It would be important to include a section describing the data analysis, treatments, and statistical metrics that were used for these specific analyses.

It would be important to include: data standardizations such as sensor frequencies for comparison with the reference, if there was a change in frequency, how the amount of valid data for the calculations was considered; a description of the calibrations or validations applied; statistical metrics used in analyzes such as RMSE, REU, etc., a simple description would add a lot to the article; Another point would be the pollutants used ($PM_{2.5}$, $NO_2$) and because these, if there are analyzes for the others, it would be interesting to mention

**SECTION 3**

Why were some analyzes used PM2.5 and others NO2? Would there be any explanation? From section 3.4 onwards, sensors are no longer specified from which manufacturer. For example, in Figure 7, which sensors are being compared? Is it just from one manufacturer? Or multiple manufacturers? Is only one sensor from each manufacturer considered or multiple?

In figure 8, what would sensors A and B be, are they from the same manufacturer or different?

Figure 10. Is the analysis for NO2? If yes, specify in the legend. The companies are unidentified, wouldn't it be possible to associate them with each one?

---

## Author Comment (AC1)

**Response to reviewers - Long-term Evaluation of Commercial Air Quality Sensors: An Overview from the QUANT Study**

Sebastian Diez, Stuart Lacy, Hugh Coe, Josefina Urquiza, Max Priestman, Michael Flynn, Nicholas Marsden, Nicholas A. Martin, Stefan Gillott, Thomas Bannan, and Pete Edwards.

We thank the referees for their time reviewing our manuscript and their useful comments and feedback. Based on the reviewers' feedback, we have made several changes which we feel significantly improve the manuscript.

Below, you will see:

- reviewer comments in **bold**
- our responses are in regular type (Calibri font).
- cited text from manuscript and supplementary in Times New Roman font.

Attached we have also provided a "track changes" version of the manuscript, with added text in blue and deleted and/or moved text in red.

**Reviewer#1**

**This study is a comprehensive, long-term study of a wide range of air sensors currently available in the market. Because of the collocation in time and space during the comparison, environmental variables are lessened thereby focusing the comparison on the performance of the air sensors relative to one another and in comparison, to a "reference" monitor. This adds great value to the study. The sensors were deployed and collocated in a real-world environment that they will most likely be used, so the conditions at which the sensors are compared were not biased because of the environment (i.e., compared to if it were collocated in a "pristine" environment, or under laboratory conditions). This study includes both gas and PM sensors which adds to the novelty of the study.**

**This paper is recommended for publication in AMT with revisions as outlined in this discussion.**

Response: Thank you for your encouraging feedback and recognition of our study's comprehensive and practical approach to evaluating air sensors.

**GENERAL COMMENTS**

**- The paper can benefit from tightening up language and being more succinct and concise in its statements.**

Response: We appreciate the feedback on the need for clearer and more concise language throughout our manuscript. To address this, we have carefully reviewed our text, focusing on simplifying complex sentences, removing redundant phrases, and ensuring that each statement directly contributes to our argument or findings.

**- A glossary of terminologies for commonly used yet widely misused or confused terms in the field (e.g., "sensor" vs "sensor systems" vs "sensing unit", "manufacturer" vs "company", "model"/"unit"/"type"/"device") may be useful to the reader, also to help guide the authors in using consistent terminology all throughout the paper.**

Response: We acknowledge the importance of clear and consistent terminology in our manuscript. We have chosen to complement the initial footnote instead of creating a glossary of terms, as we think this is a more practical use of these definitions. Please refer to section "1. Introduction" to check the usage of "sensor" and "sensor systems". As for the terms "manufacturer" and "company", please refer to the section "2. QUANT study design". This approach offers immediate contextual explanations without diverting the reader's attention from the main content. It also accommodates the diverse backgrounds of our audience by providing specific clarifications tailored to the context of this specific study. These footnotes now can be read:

[1] The term "sensor systems" refers to sensors housed within a protective case, which includes a sampling and power system, electronic hardware and software for data acquisition, analog-to-digital conversion, data processing and their transfer (Karagulian et al., 2019). Unless specified otherwise, the term "sensor" will be used as a synonym of "sensor systems". Other alternative names for "sensor systems" used here are "sensor devices" (or "devices"), "sensor units" (or "units").

[2] In a narrower sense, "sensor" typically denotes the specific component within a sensor system that detects and responds to environmental inputs, producing a corresponding output signal. To distinguish this from the broader use of "sensor" as equivalent to "sensor system" in our text, we will utilise alternative terms such as "detector", "sensing element", or "OEM" (original equipment manufacturer) when referring specifically to this component, thereby preventing confusion.

[3] Throughout this article, the terms "manufacturers" and "company" are used interchangeably to refer to entities that produce, and/or sell sensor systems or devices. This usage reflects the industry practice of referring to businesses involved in the production and distribution of technology products without distinguishing between their roles in manufacturing or sales.

As for the term "type" it has been removed from the text and replaced with "model and brand" for more specificity.

**- A major issue is the description and reliance on usual metrics like R2 in comparing two instrumental methods. With a goal of prediction and calibration in mind, R2 is an appropriate statistical metric; however, in plainly comparing the correlation (specifically: the concordance or agreement between two measurements), the authors are recommended to use more appropriate statistical metrics that measure concordance such as the concordance correlation coefficient. The paper can also greatly benefit a wider audience if the authors expanded on the statistical discussion and provide a separate discussion of the statistical metrics used, thus serving as a technical guidance that outlines metrics that can be used in such an intercomparison or calibration exercise.**

Response: Thank you for your insightful comments. We acknowledge the limitations of relying solely on metrics like R2 for comparing instrumental methods. To address this, we have previously published a paper discussing the limitations of single-value metrics and advocated for a combined approach using visualisation tools and a range of metrics for a more nuanced analysis. For an in-depth discussion on this topic, we direct readers to Diez et al., 2022. Furthermore, the lead author of this manuscript has contributed to a chapter on performance metrics in an upcoming World Meteorological Organization report (to be published in May 2024), which delves into the advantages, limitations, and best practices regarding performance metrics in greater detail.

In this overview paper, we aim to provide readers with a holistic view of sensor performance by employing a variety of metrics and visualisation techniques. While we recognize the value of the Concordance Correlation Coefficient and its relevance, we also note its limitations. No single metric can fully capture all aspects of sensor performance, prompting our choice of widely recognized metrics like R2, RMSE, and MAE. This choice aims to facilitate comparisons with previous studies and standards in scientific literature.

In response to your suggestion, we have expanded our discussion on the statistical metrics used, adding a section in the supplementary materials (section "S5. Performance Metrics") that serves as a summary discussion of the topic. Additionally, we have included a summary of recent standardisation efforts (Table S6), which we believe will be beneficial for end-users seeking guidance on metric selection for intercomparison or calibration exercises.

SPECIFIC COMMENTS

**- For the title: A more specific term than "Evaluation" can be used. Suggestions: "Intercomparison"; "Precision Analysis"**

Response: Thank you for your suggestion. We've chosen "Evaluation" as it most accurately encompasses the study's scope, which goes beyond comparison and precision analysis to include a comprehensive assessment of devices across various end-use scenarios. This term reflects our analysis's breadth, covering precision, accuracy, long-term performance, and adaptability to environmental conditions, thereby providing a holistic view of the sensors' capabilities for potential users.

**- The abstract is missing some key findings and results, e.g. how many air sensors and reference sensors were quantified, statistical metrics used to quantify the performance of the sensors, etc. For example, Lines 89-90 can be added to the abstract.**

Response: Thank you for your valuable feedback. In response, we have enriched the abstract to reflect the number of sensors and companies, as well as the range of metrics and visualisation tools utilised in our study.

**- A separate section on the methodology summarizing performance metrics used, and explaining each under a subheader, e.g., "Bias" as a subheader and explaining R2, RMSE, etc. under this heading would be useful to the reader and also makes this a good reference paper for intercomparison studies.**

Response: Thank you for your suggestion. As mentioned in response to an earlier comment, we have expanded our discussion on the metrics used. Please to section "S5. Performance Metrics" in the supplementary.

**- Section 3.3 explores reference instrumentation. Authors need to make what is meant by "reference", e.g. a reference method designated by an authority (EU, US EPA, etc.) or a self-defined or agreed-upon reference method.**

Response: Thank you for your suggestion. The text has been complemented in order to clarify this point:

For an overview of reference and equivalent-to-reference instrumentation, as defined in the European Union Air Quality Directive 2008/50/EC (hereafter referred to as EU AQ Directive), at each site, please refer to Section S2 (Table S1). For details on the quality assurance procedures applied to the reference instruments, see Table S2.

Table S1 has also been updated to reflect this.

**- In one section, the manufacturers / models of air sensors were referred to; however, in Figure 10, it was anonymized. What was the rationale? Can it be consistently anonymized or named? And if not, explain why and make sure that the transition is clearly explained within each section.**

Response: Thank you for highlighting this aspect. Our initial decision to anonymize the names of companies in certain figures, including the original version of Figure 10, aimed to center the discussion on the generalizable features of sensor technologies rather than on specific findings tied to individual manufacturers. This aimed to minimise potential biases and promote a comprehensive understanding of the technology in question. Nevertheless, acknowledging the significance of transparency and in response to constructive feedback, we have updated Figure 10 to reflect the names of the companies. Furthermore, we have elaborated on this rationale within the "3. Results and discussion" section to ensure a clear and consistent explanation of our approach to anonymization versus explicit naming:

To highlight broad implications and insights into sensor technology, rather than focusing on the performance of specific manufacturers, figures illustrating brand-specific features have been anonymized. This is intended to prevent potential bias and encourage a holistic view of the data, ensuring interpretations remain focused on general trends rather than isolated examples.

In addition, we are preparing a series of articles that will delve into more granular aspects of our study, as outlined in the conclusion section. Moreover, the dataset has been made publicly available, enabling comprehensive scrutiny of sensor performance by the broader community. A data descriptor paper detailing the QUANT dataset has also been submitted and is currently under review; once published, it will provide users with full access to and understanding of the dataset, further enhancing transparency and facilitating research in this field.

**- It is useful from a consumer perspective to mention which devices are available readily as-is (without add-ons) and/or which ones require customization from the manufacturer's end. This can possibly be added to the summary in Table 1 and/or Supplementary S1, with a short reference (a sentence or two) in the main text.**

Response: Thank you for your suggestion. The text description of tables 1 and 2 has been complemented in order to clarify this point:

**Table 1. Main QUANT devices description. The 20 units, all commercially available and ready for use as-is, offered 56 gas and 56 PM measurements in total. For a detailed description of the devices see Section S31 in the Supp.**

**Table 2. The 23 WPS devices deployed at the Manchester supersite, all commercially available and ready for use as-is, provided 63 gases and 62 PM measurements in total. For a detailed description of the devices see the Section S43 in the Supp.**

**- Might be useful to add in the conclusions / recommendations section for future researchers: quantify inter-location variability.**

Response: Thank you for your suggestion to include more recommendations in our conclusions section. In addition to our final paragraph which includes future work we intend to do to address research needs we have also added the sentence below which highlights the end-user need for impartial performance data, which researchers are in a unique position to address.

Ultimately, this work shows that sensor performance can be highly variable between different devices and end-users need to be provided with impartial performance data on characteristics such as accuracy, inter-device precision, long-term drift and calibration transferability in order to decide on the right measurement tool for their specific application.

**- Might be useful to explain and emphasize (including in the abstract) why correction with satellite data was not explored in this study.**

Response: We appreciate the reviewer's suggestion regarding the exploration of correction with satellite data. However, after careful consideration, we have decided not to include an explanation for the omission of satellite data correction in our study, both in the manuscript and the abstract. This decision is based on the focused scope of our research, which is the direct evaluation of commercial air quality sensors in urban environments. Our primary aim is to assess these sensors' performance and applicability in settings where ground-level monitoring provides the most immediate and relevant data for urban air quality assessment. Including satellite data, which typically offers broader spatial coverage but lacks the fine-scale resolution required for our analysis, would not align with the specific objectives of our study. Furthermore, we aim to maintain a concise and focused narrative that is directly relevant to our core findings and methodology. We believe this approach will serve our audience best, keeping the manuscript clear and streamlined.

**- Can employ the terms "inter-device" and "inter-location" for succinctness of ideas.**

Response: Thank you for your suggestion. We have reviewed our manuscript and incorporated these terms where appropriate to more precisely describe the variability in sensor performance across different units and sites.

**Line by line comments**

**- Line 45: "cross-sensitivity" seems to be a term usually used in the medical context. It might be beneficial to define this term in this context, and differentiate it from "interference". Levy Zamora (2022) used "cross-sensitivity" in the title of their article and Bitner (2022) defined it, so it might be helpful if these two comes as the first articles cited in this instance).**

Response: We appreciate the reviewer's input and have included Bitner (2022) for further clarification on "cross-sensitivities". However, to keep our overview concise and focused, we believe the addition of this reference, along with Cross et al. (2017) and Pang et al. (2018), adequately informs readers about sensor challenges without overextending on definitions.

**- Lines 48-49: citations for temperature and humidity might be combined since they are usually explained in cited references in combination.**

Response: Thank you for the input. We have made the change accordingly.

**- Lines 63-68: This paragraph could benefit from differentiating "calibration" from "correction" and how these two terms are sometimes interchangeably used (albeit incorrectly). A reference to an article that explains this difference will also be helpful. The Liang (2022) paper cited explains some of these nuances, including mathematical equations for calibrations, but does not fully differentiate "correction" from "calibration".**

Response: Thank you for your valuable feedback. To clarify the distinction between "calibration" and "correction", we have refined the text to include a direct reference to VIM (2012). Now the text reads:

The calibration of any instrument used to measure atmospheric composition is fundamental to guarantee their accuracy (Alam et al., 2020; Long et al., 2021; Wu et al., 2022). Using out-of-the-box sensor data without fit-for-purpose calibration can produce misleading results (Liang & Daniels, 2022). An effective calibration not only involves identifying but also compensating for estimated and correcting systematic effects errors in the sensor readings, a process defined as a correction (for a detailed definition and differentiation of calibration and correction see JCGM, 2012).

**- Lines 67-68: True for gases. Mention examples of acceptable calibration method(s) for PM?**

Response: Thank you for your suggestion. We have modified the original text in this way:

For standard air pollution measurement techniques, calibration is often performed in a controlled laboratory environment (Liang, 2021), or by sampling gas from a certified standard cylinder in the field. For PM, particles of known density and size are used, controlling the airflow conditions. For example, for gases, a known concentration is sampled from a certified standard. Similarly, for PM, particles of known density and size are generated. Both gases and PM calibration are conducted under controlled airflow condition

**- Lines 81-83: Another reason is that there are a lot of sensors/sensor systems with different configurations commercially available, and also individual sensing units are sold and can be "DIY"-ed—the market is diluted with many options and many different iterations of the same underlying technology with marginal differences.**

Response: Thank you for your suggestion. We have modified the original text in this way:

This is largely due to the significant variability in both the number of sensors and the variety of applications tested, compounded by the proliferation of commercially available sensors/sensor systems with different configurations. as well as the availability of highly accurate measurement instrumentation and/or regulatory networks to those outside of the atmospheric measurement academic field. Furthermore, the access to highly accurate measurement instrumentation and/or regulatory networks remains limited for those outside of the atmospheric measurement academic field (e.g. Lewis and Edwards (2016) and Popoola et al. (2018)).

**- Line 94: Clarify or add examples of "data products" e.g., APIs, mobile apps, etc.**

Response: Thank you for your suggestion. The revised sentence now reads:

Furthermore, we tested multiple manufacturers' data products, such as out-of-the-box data versus locally calibrated data, for a significant number of these sensors to understand the implications of local calibration.

Section "2.3 Data collection, co-located reference data and data products" has also been updated (see also response to line 158 comment, page 10).

**- Lines 105-106 and 116: Useful to add a subsection that describes the UK urban environment including seasonality, sources of pollution (transportation? Household commercial products use?) in the three locations (London, Manchester and York).**

Response: We appreciate the suggestion and recognize the importance of seasonality and pollution sources on sensor performance. However, a detailed exploration of seasonal variations at the 3 sites extends beyond this study's scope. The need for such analysis, considering the uncertainties around the UK's environmental characteristics, motivated the initiation of the Integrated Research Observation System for Clean Air (OSCA), the measurement component of which was underway during the QUANT study. OSCA's forthcoming outputs are expected to provide in-depth environmental insights, and these data will be used in future work for a more in depth study of sensor interferences etc. Nonetheless, we've updated the supplementary material to include a more detailed description on urban environment details and anthropogenic activities, for each site. Please refer to "S1. Co-location sites".

**- Line 106: "replicates" or "units" are more appropriate terms than "duplicates" if you are talking of the units of the same model**

**- Line 109: define what is meant by "near real-time" in this context.**

Response to the last two comments: Thank you for your suggestion. The text has been updated accordingly, and now reads:

Four units  of five different commercial sensor devices (Table 1) were purchased in Sept 2019 for inclusion in the study, with the selection criteria being: market penetration and/or previous performance reported in the literature, ability to measure pollutants of interest (e.g. $NO_2$, $NO$, $O_3$, and $PM_{2.5}$), and capacity to run continuously reporting high time resolution data (1-15 min data) ideally in near real-time (i.e., available within minutes of measurement) with data accessible via an API.

**- Line 113: Were the units tested together before deploying separately? Clarify.**

Response: Thank you for your comment. All the units were first deployed together in Manchester as stated in the original lines 122 and 123:

Initially, all the sensors were deployed in Manchester for approximately 3 months (mid-Dec 2019 to mid-Mar 2020) before being split up amongst the three sites (Fig. 1).

**- Line 121: A sentence or two succinctly describing the sites will also be useful in this line. Then you can refer to the Supplementary.**

Response: Thank you for your suggestion. We've slighlightly modified the current description for readers interested in more detailed information, as we have added the official web address describing each of the mentioned sites. We have also made a more thorough description of each of the sites at the supplementary (please see the response to an earlier comment regarding Lines 105-106 and 116).

**- Lines 122-126: Consider moving up before Lines 113-121.**

Response: Thank you for your suggestion. We moved the text according to the suggestion.

**- Line 125: "inter-device consistency" may also be rewriten as "precision".**

Response: Thank you for your suggestion. We acknowledge that "precision" could effectively communicate the aspect of measurement variability among devices. However, we believe "inter-device consistency" encompasses not only the precision aspect but also the reliability and stability of device measurements across various conditions and over time. Thus, we have opted for the initial term to convey the broader scope of our analysis more accurately.

**- Lines 134-136: "vendors were invited to contribute multiple sensor devices throughout the WPS study". How does a "sensor device" differ from a "sensor" or "sensor system"? Does this mean that manufacturers can contribute different sensor models? Also, does vendor = company = manufacturer? Note consistent terminology all throughout the manuscript (might be useful to have a glossary or footnote, like that for "sensor" and "sensor system" on page 2.**

Response: We appreciate the suggestion to clarify terminology. Following the initial recommendation in "General comments" (page 2), we have carefully defined and differentiated these terms as footnotes in the "Introduction" and in "QUANT study design" sections. As for the word "vendor" (it appeared two times in the original text), it was replaced by the word "manufacturer".

**- Lines 139-141: Table 1. Does AQMesh AQM, Kunak AP, and SCS Prax have all of the sensors listed (from NO to PM10) in one unit? This table might benefit from a clarification (can be added to the Table caption). Also, as mentioned in a previous comment, add in the description if these are consumer-ready (eg already sold in the market as that unit), or customizations available from the manufacturer.**

Response: Thank you for your suggestion. As stated in "Specific comments" (page 5), the text description for tables 1 and 2 has been adapted to help clarify this point.

**- Lines 148-149: I understand that PurpleAir does not have a mobile data connection, only WiFi, but WiFi was not good in the location so you opted to download the data from the device memory instead. The text can be enhanced by beter explaining the issue as described. (i.e., differentiating from WiFi and mobile data connectivity)**

Response: Thank you for your suggestion. We updated the text, and now reads:

 Unlike other brands that utilize mobile data connections, PurpleAir sensors rely on WiFi for data transmission. Due to poor internet signal at the sites, we locally collected and manually uploaded readings for these units.

**- Line 150: and harmonize? In the methodology section, it might be useful to mention that temporal and spatial scales of the sensor systems was important to match, thus aggregation and harmonization was necessary. How was incomplete data treated? Were there imputed data? Might be useful to add it in the supplementary.**

Response: Thank you for your suggestion. We've clarified in our manuscript that "standardize" includes formatting, aggregating, and ensuring data compatibility across devices, without altering data characteristics. This revision is now reflected:

Minor pre-processing was applied at this stage, including temporal harmonisation to ensure that all measurements had a minimum sampling period of 1-minute, ensuring consistency in measurement units and labels, and coercing into the same format to allow for full compatibility across sensor units.

On the other hand, incomplete data didn't receive special consideration as was originally stated. We've now expanded this in the text to avoid confusion:

No additional modifications to the original measurements were applied; missing values were kept as missing and no additional flags were created based on the measurements beyond those provided by the manufacturers.

**- Line 150: by data format, do you mean datetime / time and date?**

Response: Thank you for your inquiry. As per the response to the previous question, the phrase "data format" has been replaced by an explanation of the specific pre-processing steps applied.

**- Line 158: "calibrated data products": is this referring to API? Measurements? As with my previous comment – clarify what "data products" mean.**

Response: Thank you for your comment. As described in our manuscript, this term refers to the various versions of data provided by manufacturers, reflecting different stages of calibration and adjustment based on colocated reference data. We believe this description, along with the supplementary material, offers a comprehensive insight into what is encompassed by this term. However, we have decided to complement the text, for clarity:

However, those who did were expected to create and submit calibrated data products, subsequently named as "out-of-box" (initial data product), "cal1" (first calibrated product), and "cal2" (second calibrated product). This differentiation highlighted the varying degrees of engagement and application of the reference data by different manufacturers. Figures S2 and S3 (section S3 and S4 respectively) show a time-line of the different data products.

**- Lines 160-166: What is cal1? cal2? Clearly define / describe these in the text and/or supplementary. This section may benefit from a subsection explaining / describing these.**

Response: see previous response.

**- Lines 170-174: Is this a caveat / weakness of using these statistical metrics used herein (R2, MAE, etc)? What is the alternative? I suggest concordance (agreement) metrics, such as the Concordance Correlation Coefficient: See Lin, Biometrics (1989): https://doi.org/10.2307/2532051. The reader might also benefit from a separate subsection and/or supplementary section describing the metrics or including a glossary of the metrics used.**

Response: Thank you for your comment. While we agree that the Concordance Correlation Coefficient is a valuable metric for assessing inter-rater agreement, it still suffers from the same limitation as described in the text, wherein over-reliance upon a single metric can obscure the full picture. Instead, we opt for a more holistic assessment comprising multiple facets of a sensor's performance. Also, we've focused on more commonly used metrics within both the scientific community and technical guidelines for sensor evaluation. This choice aims to provide a comprehensive assessment of sensor performance and facilitate comparisons with existing guidelines and research findings. We've updated the text to better convey this:

 Furthermore, the overreliance on global performance metrics is a significant concern in sensor assessment. The Coefficient of Determination ($R^2$), Root Mean Squared Error (RMSE), and Mean Absolute Error (MAE) are among the most popular single-value metrics for evaluating sensor performance, alongside others (e.g., the bias, the slope and intercept of the regression fit). However, while single-value metrics offer an overview of performance, they can be limiting or misleading. They condense vast amounts of data into a single value, simplifying complexity at the expense of a nuanced understanding of error structures and information content (Diez et al., 2022), potentially overlooking critical aspects of sensor performance (Chai & Draxler, 2014). Visualisation tools (such as Regression plots, Target plots, and Relative Expanded Uncertainty plots) complement these metrics, allowing end users to identify relevant features, which could be beyond the scope of global metrics. For further discussion on metrics and visualisation tools for performance evaluation, readers are directed to Diez et al. (2022).

As for the suggestion section describing the used metrics, please refer to response to previous suggestions (page 3).

**- Line 183: "multiple devices of the same type" when you mean "type" do you mean similar underlying principles of measurement? Model? Be consistent in terminology. Also, it might be useful to cite an example of which devices you are considering a same "type", e.g. AQM and Clarity—are these of the same "type" as described?**

Response: Thank you for your suggestion. As stated in a previous response from "General comments" (see page 2), the term "type" has been replaced with "model and brand" for more specificity.

**- Lines 190-196. See also deSouza (2023): An analysis of degradation in low-cost particulate mater sensors htps://doi.org/10.1039/D2EA00142J**

Response: Thank you for your suggestion.

**- Lines 202-204: Good point.**

Response: Thanks!

**- Line: 217: 75% inclusion criteria is common—but perhaps not for readers not familiar with this data type. Readers might benefit from a citation, explanation in methodology or supplementary. Suggested section to add it in: Section 2.1, lines 150-151.**

Response: Thank you for your suggestion. To address the comment and enhance clarity, we have specifically mentioned the data inclusion criterion of 75% at the end of the last paragraph in the "3. Results and discussion" section:

The following sections aim to provide an overview of the data and provide initial findings, with a focus on those that are most relevant to end-users of these technologies. All metrics and plots presented here are based on 1-hour averaged data. Unless otherwise specified, a data inclusion criterion of 75% was uniformly applied across our analyses to ensure the reliability and representativeness of the results. This threshold aligns with the EU AQ Directive, which mandates this proportion when aggregating air quality data and calculating statistical parameters.

**- Line 225: Clarify: did you mean closer together spatially / physical location? How "close" is close?**

Response: Thank you for seeking clarification. "Closer together" refers to the clustering of sensor performance data points in the plot, indicating improved measurement consistency among sensors from the same manufacturer after calibration. We have updated the text to avoid this confusion:

Secondly, it can help to improve within-manufacturer precision , as evidenced by sensor systems from the same company grouping more closely as the right plot in Fig. 4 shows.

**- Line 232: could benefit more from a further explanation of the bias-variance tradeoff.**

Response: Thank you for the suggestion. We have updated the text to be more descriptive.

For the out-of-the-box data, these regions are noticeably larger than in the calibrated results for most manufacturers, suggesting that colocation calibration has helped to tailor the response of each device to the specific site conditions.  This observation suggests that colocation calibration effectively improves each device's response to particular site conditions. This improvement is underscored by the more substantial reduction in the cRMSE component compared to the MBE. The cRMSE, representing the portion of error that persists after bias removal, essentially measures errors attributable to variance within the data space. In the context of out-of-the-box data, this "data space" spans all potential deployment locations used by manufacturers for initial calibration model training (i.e., before shipping the sensors for the QUANT study), thus exhibiting high variability. However, applying site-specific calibration significantly narrows this variability, leveraging local training data to minimise variance.

**- Section [3.3. and Supplementary S4. Cite the authorities that consider the instruments mentioned as reference. e.g., are they considered "reference" because they are listed in an EU directive or US EPA documentation? If so, cite these. If not, provide a rationale or a citation as to how these instruments were categorized as "reference".**

Response: Thank you for your suggestion. This point was already addressed in the "Specific comments" (refer to page 3).

**- Lines 241-242: Expound on the significance and advantages of REU as opposed to the other metrics.**

Response: Thank you for the suggestion. We have updated the text accordingly:

For applications where it is important to understand how calibrations impact lower or higher percentiles, considering other metrics or visual tools would be advisable. An example of this is the absolute and Relative

Expanded Uncertainty (REU, defined by the Technical Specification CEN/TS 17660-1:202). Unlike the more commonly used metrics such as $R^2$, RMSE, and MAE, which measure performance of the entire dataset, the REU offers a unique "point by point" evaluation, enabling its representation in various graphical forms, such as time series or concentration space (for the REU mathematical derivation, refer to section "S5. Performance Metrics"). The REU approach, also incorporates the uncertainty of the reference method into its assessment, highlighting the intrinsic uncertainty present in all measurements, including those from reference instruments. This consideration of reference uncertainty is crucial for a holistic understanding of sensor performance and calibration effectiveness. For a comprehensive discussion on this, refer to Diez et al. (2022).

**- Lines 268-269: clarify / reiterate the said minimum requirements in this text.**

Response: Thanks for this request. The modified text now reads:

The REU demonstrates that, under these circumstances, an instrument designated as a reference does not meet the minimum requirements (REU ≤ 15%) set out by the Data Quality Objectives (DQOs) of the EU AQ  Directive .

**- Lines 283-284: R2 can many times be subjective, e.g. how can we say that an R2 of 0.87 means "it does not fully agree" and a slope of 0.80 is considered a pronounced bias? This is where the definition of concordance and using concordance metrics might be useful. If two measurements are concordant (in agreement), then slope is expected to be unity (=1). Also, a high R2 does not necessarily mean agreement between the two instruments. Also, clarify what is meant by "limiting the linearity". The authors are cautioned against using R2 in quantifying agreement between two instruments that are being compared.**

Response: We appreciate the feedback and understand the nuances involved in interpreting these metrics within the specific context of air quality studies. The interpretation of any single-value metric, like R2, RMSE, MAE —including the CCC— inevitably involves a degree of subjectivity, relying on the analyst's expertise to discern their significance within the context of the study. These statistical measures compress a vast amount of data into a singular value, potentially obscuring the broader picture. In our study we opted for a holistic assessment trying to encompass multiple facets of a sensor's performance, integrating both quantitative metrics and visual analyses to offer a comprehensive evaluation rather than placing sole emphasis on any single metric

Regarding the interpretation of an R2 value of 0.87, we welcome the correction that R2 doesn't directly measure the degree of agreement between 2 sensors and have updated the text to be more precise by referring to the linearity. We have described this as a "strong association", which albeit a subjective term, is justified by a Pearson's R > 0.90, which is in-line with conventional usage (for example https://pubs.acs.org/doi/10.1021/acsearthspacechem.8b00079).

In response to the comment regarding the slope of 0.80, it appears there may have been a misunderstanding. Our original manuscript stated that such a slope "is not considered a very pronounced bias". We value this opportunity to clarify our intent and have refined our wording for greater clarity and to avoid any potential ambiguity. The modified text reads:

To illustrate these differences in practice, Fig. 6 compares these two equivalent-to-reference PM$_{2.5}$ measurements obtained with a BAM (AURN York site, located on a busy avenue), and a FIDAS unit specifically installed for QUANT. During this specific period, they show a strong linear association  (R$^2$ = 0.87). Although the bias is not extremely pronounced (slope=0.80), the FIDAS measurements are, on average, systematically lower compared to BAM.

**- Lines 288-289: Specify criterion stipulated by EU DQOs**

Response: Thanks for this request. The modified text now reads:

In the hypothetical case that the BAM were to be considered the reference method (arbitrarily chosen for this example as it is the current instrument at the AURN York site) when assessing the FIDAS under these test conditions, it would only meet the criterion stipulated by the EU DQOs for indicative measurements (REU ≤ 25% for PM$_{2.5}$), but not for fixed (i.e., reference) measurements (REU ≤ 50% for PM$_{2.5}$).

**- Line 308: "reasonably consistent" – reasonably is subjective and qualitative. Suggest dropping the word, or provide a percentage of the time that the RMSE is consistent (e.g. provide a qualitative measure).**

Response: Thank you for highlighting this. In response to your suggestion, we have clarified the statement and the revised text now reads:

The RMSE remains reasonably consistent (range 2.27 to 3.47 ppb) between the devices across the periods and locations

**- Line 312: "local conditions": give or name some examples. Do you mean weather conditions? Traffic? etc.**

Response: Thanks for this request. The modified text now reads:

The precise cause of this change is not immediately evident and will be the focus of a follow-up study, but could be due to changes in local conditions (e.g., weather, emissions, etc.) impacting sensor calibration and/or differences in actual PM$_{2.5}$ sources and particle characteristics at the sites (Raheja et al., 2022).

**- Line 334: quickly define (add a phrase) that describes "overfiting"**

Response: Thank you for your suggestion. Accordingly, the text has been revised to include a brief definition of overfitting:

Furthermore, it is important to consider that excessive post-processing may lead to overfitting —a situation where a model excessively conforms to specific patterns in the training data, resulting in poor performance on new, unseen data (Aula et al., 2022).

**- Line 336: "linear correction" – "linear regression" might be the appropriate term.**

Response: Thank you for your suggestion. In the context of our study, "linear correction" was deliberately chosen to describe the application of a simple linear adjustment to sensor data (e.g., zero and span correction).

**- Line 347: RMSE also showed seasonality.**

Response: Thank you for your comment. You're correct that RMSE exhibits some seasonality. Our analysis primarily aimed to emphasise the behaviour of its orthogonal components: MBE and cRMSE (i.e., RMSE is as a function or consequence the latter). We have slightly modified the figure caption in order to acknowledge this point:

**Figure 9. Seasonal variation of error (as RMSE, red line) of one of the systems belonging to the Main QUANT…**

**- Lines 345-351: Add more explanations about the seasonality. Add recommendations.**

Response: Thank you for your valuable feedback. We believe that the existing sections of this overview paper, particularly 3.5 and 3.6, already offer a detailed discussion on the temporal nature of sensor errors, including seasonality aspects (lines 345-351), and outline practical recommendations for improving sensor data quality, highlighting methods like NO2 bias correction using diffusion tubes (lines 388-396). Recognizing the importance of a deeper investigation, our manuscript also outlines future studies (lines 463-466) dedicated to a more thorough exploration of seasonality effects and the development of detailed recommendations. This is part of our broader commitment to improving the understanding and application of sensor performance, with successive studies planned to delve into these aspects further.

**- Line 355: Clarify what a "1-day slide" means – it can be added in the supplementary or a quick description in the figure caption.**

Response: Thank you for your suggestion. We have updated the description in the figure caption, and now reads:

**Figure 9. Error (as RMSE, red line) of one of the systems belonging to the Main QUANT, decomposed into cRMSE (in blue) and MBE (in yellow) estimated based on a 40-day (aligning with the sample size recommendation by the CEN/TS 17660-1:2021 standard for on-field tests) moving window approach with a 1-day slide (i.e., advancing the calculation 1 day at a time) (1-day slide) moving window. Panel a) is for O$_3$ measurements, and panel b) is for NO$_2$ (April 2020-Oct 2022). Panel c) is also for NO$_2$, this time showing the effect of a linear correction using diffusion tubes (see next section for more details).**

**- Lines 363-366: This can be said more succinctly. Also, what sort of information can be provided? Be specific, based on your results so far—what sort of information can you recommend be provided?**

Response: Thank you for your feedback. In response, we have revised this section to make it more concise and to explicitly outline the type of information that should be provided. The revised text now reads:

In order to realise the potential of air pollution sensor technologies, end users need to be provided with the information required to critically assess the strengths and weaknesses of potential candidate sensor devices, ideally in an easy to access and interpret manner. To realise the potential of air pollution sensor technologies, end users need to align their specific measurement needs with the capabilities of available devices. Achieving this necessitates access to unbiased performance data, such as long-term stability and accuracy across varying conditions, ideally in an easy-to-access and interpret manner.

**- Line 373: Inconsistency in the use of the term "sensor", "system" and "sensor system".**

Response: See previous response on "General comments" (see page 2).

**- Lines 377-379: This discussion can benefit from a more detailed explanation of the tiers (Classes) assigned and what was the basis of the assignment to different classes. The figure caption for Figure 10 offers an explanation, which should be repeated and explained in more detail in-text.**

Response: Thank you for your suggestion. While we value the importance of detailed explanations regarding the classification into different tiers, we aimed to provide a broad overview within the scope of this manuscript, directing readers to specific documents (i.e., EU AQ Directive and CEN/TS 17660-1:2021) for comprehensive details on class assignments and the rationale behind them. This approach ensures that our manuscript remains focused on providing an accessible overview of the QUANT study, while still making in-depth information readily available to those interested in delving deeper. We have however modified this section, and now reads:

Both REU and DC are key criteria within the EU scheme (EU 2008/50/EC) for evaluating the performance of measurement methods, and are complemented by the CEN/TS 17660-1:2021 specifically for sensors. The latter  document defines three different sensor system tiers. Class 1 $NO_2$ sensors, bounded by the green rectangle $(REU < 25\%$ and $DC > 90\%)$, offer higher accuracy than Class 2 sensors $(REU < 75\%$ and $DC > 50\%)$, delimited  by the red rectangle (Class 3 sensors have no set requirements). Presenting the REU and DC  like in Fig. 10  helps users anticipate the performance of sensor systems —under the assumption that all sensors from the same brand will behave similarly in equivalent environmental conditions— providing more insight into selecting the appropriate instrument for a given project or study.

**- Line 388 onwards can benefit from a separate subheader / subsection.**

Response: Thank you for your suggestion. After careful consideration, we would prefer to maintain the current structure of Section 3.6. As the intention of this overview paper is to summarise the study and some of the key findings as well as point potentially interested readers towards our unique dataset, we are keen to avoid it being overly exhaustive and long. We also feel that our manuscript has thematic coherence, as Section 3.6 transitions from discussing sensor performance nuances to practical implications for end-use applications. The seamless flow into "Informing end-use applications" is intentional, reflecting our comprehensive approach to presenting both technical analysis and its practical implications in a unified narrative.

**- Lines 390-391: Specify an example of "simpler methods"**

Response: Thank you for highlighting the need for clarification. We intended to convey that, depending on the application, there might be other feasible alternative methods for bias, rather than "simpler" from a technical point of view. To clarify this in our manuscript, we have refined the text as follows:

 Depending on the application and available options, users can access alternative methods to reduce bias, thus enhancing the accuracy of sensor systems and networks. For example, "Indicative methods", as defined by the EU AQ Directive, such as diffusion tubes (e.g., NOx, SO₂, VOCs, etc.), can be an option. Specifically, our study leverages diffusion tube data for NO₂, illustrating one effective approach to bias correction using supporting observations, as exemplified in Fig. 9b.

**- Line 393: explain more by what instrumental method is NO2 measured/detected from these diffusion tubes. Cite a reference as well.**

Response: Thank you for raising this question. We would like to clarify that a detailed explanation of the instrumental method used for NO2 detection via diffusion tubes was already included in the supplementary material of our original submission, specifically in the section "S6. NO2 Diffusion Tubes" (renamed after revisions as "S7…")

**- Line 411: "change points": does this mean inflection points? Periods of biggest slopes? Peak concentration periods? Consider changing verbiage.**

Response: Thank you for your suggestion. To clarify, we've added a brief description of change points within the statistical field of change detection to our manuscript:

An example of this from the QUANT dataset is the use of sensor devices to successfully identify change points in a pollutant's concentration profile. These are points in time where the parameters governing the data generation process are identified to change, commonly the mean or variance, and can arise from human-made or natural phenomena (Aminikhanghahi and Cook, 2017).

**- Lines 414-415: Is it applied in this paper? If so, this line should be described in the methodology and explained further.**

Response: Thank you for your comment. The mention of change point analysis was intended to illustrate the potential applications of the QUANT dataset, rather than to detail a methodology applied within this specific study. To clarify and prevent any misunderstanding, we have revised our manuscript, and now reads:

Determining when a specific pollutant has changed its temporal nature is a challenging task as there are a large number of confounding factors that influence atmospheric concentrations , including but not limited to seasonal factors, environmental conditions (both natural and arising from human behaviour), and meteorological factors. This challenge has lead to several "deweathering" techniques being proposed in the literature (Carslaw et al., 2007; Grange and Carslaw, 2019; Ropkins et al., 2022). While change point detection is highlighted here as a promising application of sensor data, it represents just one of many potential methodologies that could be explored with the QUANT dataset.

We have also updated the text explaining the methodology applied to the QUANT dataset:

A state-space based deweathering model was applied to $NO_2$ concentrations measured from the sensor systems that had remained in Manchester throughout 2020 to remove these confounding factors, with the overarching objective to identify whether the well-documented reduction in ambient $NO_2$ concentrations due to changes in travel patterns associated with COVID-19 restrictions could be observed in the low-cost sensor systems. To provide a quantifiable measure of whether a meaningful reduction had occurred, the Bayesian online change-point detection (Adams & MacKay, 2007) was applied. Of the 8 devices that measured $NO_2$, clear change points corresponding to the introduction of a lockdown were identified in 2 (Fig.11), demonstrating the potential of these devices to identify long-term trends with appropriate processing, even with only 3 months of training data.

**- Line 421: Expound what "unsupervised analysis" means in this context. General verbiage related to machine learning may sometimes be unnecessary to use in this text, and can be avoided, because fundamental/rudimentary statistical metrics (as opposed to complex "black-box" machine learning algorithms) are used.**

Response: Thank you for your comment and suggestion. In this context, "unsupervised analysis" refers to the application of statistical techniques without explicit guidance or labelled data. For this case, it means without directly comparing the modelled output (estimated change-point) to the actual measured outcome (e.g., date of Covid lockdown). Acknowledging this can cause misleading interpretations, this term has been removed with the text.

**- Line 422: consistency in terminology. Do the authors mean "sensor system" when they mention "devices"?**

Response: See previous response on "General comments"

**- Line 433: "..use of these devices has been primarily limited…" I would disagree, because consumers and many users still use these devices (sensor systems) and they aren't necessarily limited by accuracy concerns, e.g. many users are willing to accept a large margin of error for awareness purposes.**

Response: Thank you for your perspective. We acknowledge that despite concerns over data quality, there is a significant user base that utilises these sensor systems for various purposes, including general air quality awareness. We had intended the limitations mentioned in line 433 to be contextualised by the preceding statement about the potential of low-cost sensors to enhance air pollution management and understanding, but have edited the sentence to provide clarity. This now reads:

Large-scale uptake in the use of these devices for air quality management has, however, been primarily limited by concerns over data quality and a general lack of a realistic characterisation of the measurement uncertainties making it difficult to design end uses that make the most of the data information content.

**- Line 439: suggested addition: (limitations in) technical ability in post-processing of data**

Response: Thank you for the suggestion. To clarify this point, we have modified the text in this way:

A challenge with the use of sensor-based devices is that many of the end-use communities do not have access to extensive reference-grade air pollution measurement capability (Lewis & Edwards, 2016), or in many cases, expertise in making atmospheric measurements or the technical ability for data post-processing.

**- Lines 460-461: Will this be done by the authors in a future study, or is this a call/recommendation for other researchers?**

Response: Thank you for your inquiry. The future studies mentioned in the concluding paragraphs of our manuscript (lines 460-469) are currently being undertaken by our team and will be detailed in forthcoming publications.

**- Line 466: suggestion for a future study: explore different VOC-NOx regimes (see Wennberg, ES&T Air: https://doi.org/10.1021/acs air.3c00055)**

Response: Thanks for your suggestion!

**TECHNICAL CORRECTIONS**

**Grammatical, Typographical, Figure and Formating comments throughout the text**

**- Note the usage of "data" as a plural noun, e.g. "data were" rather than "data was"**

Response: Thank you for pointing this out. This is now corrected.

**- "Manufacturer" rather than "Company" might be a more descriptive noun for the intended usage.**

Response: Thank you for your suggestion. In a previous comment, we've already addressed this concern.

**- "co-location" vs "collocation"? Stay consistent.**

Response: Thank you for your observation. We've addressed this by ensuring the usage of "co-location" throughout the text.

**- Many links in the "References" section of the supplementary point to a Zotero page that is meant for Google docs, thus rendering the links inaccessible**

Response: Thank you for bringing this to our attention. We have revised the links in the "References" section to ensure accessibility and functionality.

**Figure comments**

**In general, labeling the figure panels with leters (e.g. (a), (b), (c), (d)) allows for easier and clearer reference in text and in figure captions. (e.g. Line 327 mentions the "top row" in Figure 8)**

Response: While we appreciate the recommendation, we believe that maintaining the current status is appropriate to avoid potential visual clutter. We will leave the decision to the editor's discretion.

**- Figure 1. Good visual—a nice representation of the timeline of events.**

Response: Thank you for the compliment; we appreciate your positive feedback. In the reviewed draft, we have also slightly enhanced Figure 1 by explicitly adding important dates to the study's timeline, as well as the names of the companies and the number of systems involved.

**- Figure 7. Which sensors are being compared here? Why the anonymity compared to the other section(s)? Also, the readers may benefit from a colorblind-friendly and more contrasting color palete. "Class 1" and "Class 2" sensors are not actually described until page 15 (line 377 onwards) – it might be useful to refer to this section (i.e. Section 3.6) in the figure caption or the accompanying text (paragraphs) that describes this figure, and mention that it will be thoroughly explained in that section.**

Response: We sincerely appreciate the suggestion for clarification. Our decision to anonymize sensors in specific figures intentionally focuses our analysis on evaluating broader sensor technology rather than individual brands. This approach aims to prevent biased interpretations, encouraging a general understanding of technological capabilities and limitations. We elaborate on our reasoning for anonymization in the "3. Results and discussion" section (see the earlier response to this point).

Following Copernicus guidelines, we ensured Figure 7 is accessible to all readers, including those with CVD, by adopting a colorblind-friendly palette (using Python's seaborn library "colorblind" option). We further validated the figure's colours via the Color Blindness Simulator (https://www.color-blindness.com/coblis-color-blindness-simulator/), as per Copernicus guidelines.

Regarding the figure's initial oversight in referring to "Class 1" and "Class 2" sensors for PM2.5, we acknowledge that the CEN/TS 17660-1:2021 standard applies only to gases. This error has been corrected to include the Data Quality Objectives (DQOs) of the EU AQ Directive, serving here only as a reference. Consequently, the figure caption has been updated to read:

Figure 7. Regression (top) and REU (bottom) plots showing data from four PM$_{2.5}$ sensors (same manufacturer) over 2 time periods: Apr-Jun 2022 and Aug-Oct 2022. The four devices were in separate locations in the first period, but all deployed in Manchester in the second. Only for reference, we have included the PM2.5 DQOs as outlined by the EU AQ Directive (for "fixed" PM2.5 measurements, REU < 25%; for "indicative" PM2.5 measurements, REU < 50%) as horizontal dashed lines.

**- Figures 8 and 10. Explain the colorations, e.g. is it meant to be a heat map? What do the specific colors mean? Figure 10 may also benefit from higher contrasting – difficult to see the contrast especially in the lower left panel, and when the plots are printed. Dashing is also difficult to see—might benefit from greater color contrast.**

Response: Thank you for your constructive feedback. The colour gradients in Figure 10 are indeed representative of a heat map, where darker colours indicate higher densities of sensor readings within the specified REU and DC values. We have modified this figure adjusting the contrast. Additionally, the dashed lines have been thickened and their transparency reduced. As for Figure 8, the colour gradient indicates data point density, with darker colours representing lower densities and brighter colours highlighting higher densities.

**Line by Line**

**- Line 80: Suggestion: "academia" or "academic research" instead of "academic arena".**

Response: Thank you for the suggestion. We have decided to keep the term "academic arena" to broadly encompass the variety of scholarly activities related to this topic.

**- Line 104: Suggestion: reword "transparent". Suggested synonyms: open, comprehensive (this changes the meaning a bit)**

Response: Thank you for the suggestion. We have decided to keep "transparent" as it precisely conveys our intended meaning, in that all methodologies and assessment criteria are open. Much of the performance data used by manufacturers to advertise sensor devices is not transparent and thus is difficult to extrapolate to end-user applications

**- Line 118: Typo: "influenced"**

Response: Thank you for catching that typo. It has been corrected to "influenced".

**- Line 155: Suggestion: reword "ratified" to "validated"**

Response: Thank you for your suggestion. We have opted to maintain "ratified" as it is the specific terminology used by the National Physical Laboratory (NPL) in this context.

**- Line 160: is "time-line" the correct term? Perhaps "comparison" or "matrix" would be more apt for Figure S1; Figure S2 is a scater plot or a bivariate plot.**

Response: Thank you for the suggestion. We confirm that "time-line" is the correct term, as both figures Figure S1 & S2 (now renamed as Figure S2 & S3) illustrate chronological sequences.

**- Line 162: change "to use this data" to "to use these data"**

Response: Thank you for pointing this out. We have corrected the phrase to "to use these data" to adhere to the grammatical convention.

**- Line 206: "Mean Bias Error" rather than "Mean Error Bias"**

Response: Thank you for your correction, the term has been updated.

**- Line 209: enclose "out-of-box" in quotation marks; typically "out-of-the-box"**

Response: Thank you for your suggestion. The term "out-of-box" is used in this context as an abbreviated form of "out-of-the-box", facilitating its encoding within our documentation, data processing (see "Data collection") and our metadata.

**-Line 231: semi-colon after "MBE", comma after "machine learning"**

Response: Thank you for the suggestions. We have implemented the suggested changes.

**- Line 257: actual "metrological" as in measurements and units, or "meteorological" as in RH and Temp?**

Response: Thank you for your inquiry. The term "metrological" is indeed correct in this context, reflecting the focus of our discussion on sensor data uncertainty.

**- Line 271: "…hypothetical scenario where it…" does "it" refer to T200U? T500?**

Response: Thank you for your comment. The reference to "it" pertains to the T200U, as contextually established in the preceding sentences.

**- Line 275: "All of this" to "all of these"**

Response: Thank you for the suggestion. Upon review, we find the original phrasing "All of this" accurately encompasses the list of required actions as a collective process, and therefore we would prefer to retain the original wording.

**- Line 276: Add comma after "monitoring"**

Response: Thank you for your attention to detail. The change has been made.

**- Line 278: "equivalent-to-reference" – consistency in hyphenation**

Response: Thank you for your comment. The use of "equivalent to reference" (without hyphenation) within quotation marks is deliberate to signify direct terminology as specified in the EU Air Quality Directive. This precise phrasing is retained to reflect the source accurately.

**- Line 282: "obtained with a BAM at the AURN York site, located on a busy avenue" – delete parentheses**

Response: Thank you for the comment. We believe the current use of parentheses enhances the reader's understanding. Therefore, we have chosen to retain it as is.

**- Line 289: Omit "of course"**

Response: Thank you for your suggestion. We have removed "of course" from the text.

**- Line 299: capitalize "FIDAS"**

Response: Thank you for your suggestion. The term "Fidas" is presented in a manner consistent with certain source materials (including the instrument manufacturer website) and common usage within our document. Thus, we have decided to keep it in the text.

**- Lines 300-301: the choice of the reference measurement**

Response: Thanks for the suggestion. "reference method" aligns with our use of PM instruments using different measurement methods/techniques. We'd therefore like to keep the original wording for consistency.

**- Line 309: Paraphrase "saw its slope change". Suggested: …"a slope change from 0.69 to 0.86 was observed…"**

Response: Thank you for your suggestion. We have made the adjustment as suggested.

**- Line 310: change "when" to "while"**

Response: Thank you for your suggestion. We agree with your recommendation and have updated the manuscript accordingly.

**- Line 321: "despite" might not be the correct conjunction here.**

Response: Thank you for your comment. After review, we believe it accurately conveys the intended contrast, so we have decided to retain it.

**- Line 331: change "akin to this later" to "akin to the later"**

Response: Thank you for your suggestion. The correction has been applied as suggested.

**- Line 268: Redundant. Change "with a measurement instrument" to "with an instrument"**

Response: Thank you for your recommendation. We have updated the text.

**- Lines 368-369: Can be paraphrased to be more succinct.**

Response: Thank you for your suggestion. We have opted to keep the original phrasing, as it precisely communicates the critical concept of uncertainty in measurement instruments and their implications.

**- Line 383: "4-system" rather than "4 systems companies"**

Response: Thank you for your suggestion. We have decided to retain the original wording as it accurately reflects our analysis of data from the sensor systems provided by four distinct companies. Each plot in Figure 10 represents the aggregated data from all operative sensors of each of the shown companies, making "4 systems companies" the most precise description of our evaluation.

**- Line 386: add "dashed", i.e., green dashed rectangle**

Response: Thank you for your suggestion. It was corrected.

**- Line 398-399. "high time-resolution" (note hyphen placement)**

Response: Thank you for your suggestion. It was corrected.

**- Line 400: subscript on NO2**

Response: Thank you for your suggestion. It was corrected.

**- Line 400: Is "DEFRA" all capitalized, or is it "Defra" as mentioned in the acknowledgement (Line 489)?**

Response: Thank you for your pointing this out. We have chosen to retain "Defra".

**- Line 407: Consider using a different word from "digestible"**

Response: Thank you for your feedback on this term. We replaced "digestible" by "accessible".

**- Line 433: change "uptake" to "uptick"**

Response: Thank you for your suggestion. We believe that "uptake" is more appropriate in this context as it is a well-established term commonly used to describe the widespread adoption or acceptance of new technologies or practices. Therefore, we have decided to retain it.

**- Line 452: "high level" seems unnecessary.**

Response: Thank you for your observation. We believe this term is necessary to accurately convey the depth of the dataset analysis conducted.

**- Line 455: "accuracy with respect to reference methods"**

Response: Thank you for your suggestion. We believe that the current wording effectively conveys the ideas.

**- Line 471: Lacks the link to supplementary information (online version link is accessible).**

Response: Thank you for your observation. Including the link to supplementary information is indeed part of the editorial process.

**Reviewer#2**

**General Comments**

**Overall this paper provides a good overview of the QUANT study and some salient results. A few clarifications are needed, as outlined below.**

Response: Thank you for your positive feedback on the paper and for acknowledging the overview it provides of the QUANT study along with its key findings.

**Section 2.3 should describe any harmonization of the data from the sensors' reporting frequencies to a standard frequency, i.e., what was the common time frequency for which the measurements were averaged for analysis and comparison with the reference? Or was this done differently for the native reporting frequencies of each instrument? Finally, in the available QUANT dataset, are the measurements reported at the initial sampling frequency or at the down-averaged frequency (or both)?**

Response: Thank you for your inquiries regarding the handling of sensor data frequencies.

-In regards to the data harmonization, we have updated the methodology text ("2.3 Sensor deployment and data collection"), and now reads: Minor pre-processing was applied at this stage, including temporal harmonisation to ensure that all measurements had a minimum sampling period of 1-minute, ensuring consistency in measurement units and labels, and coercing into the same format to allow for full compatibility across sensor units.

-as for the data collection frequencies, we have added the following text in order to clarify this (see "2.3 Sensor deployment and data collection"):

For an overview of the sensor measurands and their corresponding data time resolutions as provided by the companies participating in the Main QUANT study and the WPS, please see Seccion S3 and S4 (Table S4 and S5) respectively.

-Regarding the analysis showcased in this overview, we processed the sensor data into hourly averages. We have added the following text to clarify this (see the "Results" section):

All metrics and plots presented here are based on 1-hour averaged data.

-The QUANT dataset reports data at 1-min time resolution. We have recently submitted a detailed manuscript that delves into the QUANT database (still under review). For more details, please refer to the response to this reviewer's last General Comment response.

**In Section 3.1, results are only presented for the PM2.5 data. I would suggest that information on the inter-sensor precision for all measurands should be provided, maybe as part of the supplemental materials, since this is a basic feature of the different sensors which can inform all the other results presented later.**

Response: While the primary aim of this manuscript is to serve as an overview —introducing the methodology used in the QUANT study, showcasing the data's potential, and highlighting broader findings— to align with this feedback we have added NO2 and O3 inter-sensor precision plots to the supplemental materials. It's important to clarify that subsequent publications will delve into detailed analyses, where more specific findings will be explored extensively.

**Since one of the goals of this paper is to introduce the QUANT dataset as a public resource for long-term performance assessment, it may be worth adding a section which details the dataset itself, or expanding the "Data Availability" section to do this. Some points to consider for this section would be the size of the dataset, the parameters included, what quality controls are applied (especially to the reference data), and any licensing of the dataset or policies associated with its use. Currently, the link provided in the "Data Availability" section does not seem to be working; presumably this will be active by the time of publication.**

Response: Thank you for your valuable suggestions. Our manuscript is primarily an overview intended to introduce the QUANT study's methodology, showcase the collected data's potential, and present general findings, rather than a detailed dataset description. The dataset's complexity, including multiple calibration products for each measured species for certain devices, made a comprehensive description challenging within this paper's scope. However, to thoroughly address the dataset specifics, we recently submitted a detailed data descriptor manuscript, providing extensive details on the collection, processing, accessibility, and structure of the QUANT dataset, including variables, reporting frequencies, and QA/QC measures. This manuscript is currently under review, and we believe it will greatly aid in understanding and using the QUANT dataset upon publication. Complementarily, we have updated the "Data Availability" text, and now reads:

The QUANT dataset, accessible at the Centre for Environmental Data Analysis (CEDA) (Lacy et al., 2023; https://catalogue.ceda.ac.uk/uuid/ae1df3ef736f4248927984b7aa079d2e), is the most extensive collection to date assessing air pollution sensors' performance in UK urban settings. It encompasses gas and PM sensor data recorded in the native reporting frequency of each device. The reference data from the three monitoring sites can be found at:

- MAQS: https://data.ceda.ac.uk/badc/osca/data/manchester;
- LAQS: https://www.londonair.org.uk/london/asp/datadownload.asp);
- YoFi: https://uk-air.defra.gov.uk/data/data_selector.

A comprehensive data descriptor manuscript, detailing the QUANT dataset's collection methods, processing protocols, accessibility features, and overall structure—including variables, data reporting frequencies, and QA/QC practices—has been submitted for publication. At the time of this writing, the manuscript is still under review.

A GitHub repository at https://github.com/wacl-york/quant-air-pollution-measurement-errors provides access to Python and R scripts designed for generating diagnostic visuals and metrics related to the QUANT study, along with sample analyses using the QUANT dataset.

**Specific Comments**

**Line 19: suggest clarification that this technology is providing the first steps for regions without pre-existing monitoring.**

Response: We appreciate your suggestion. We have revised it as follows:

In times of growing concern about the impacts of air pollution across the globe, lower-cost sensor technology is giving the first steps in helping to enhance our understanding and ability to manage air quality issues, particularly in regions without established monitoring networks.

**Line 34: "end-users" should be "end-user".**

Response: Thank you for your suggestion. Correction made.

**Line 35: "capabilities the" should be "capabilities, the".**

Response: Thank you for your suggestion. The wording was corrected.

**Line 54: "helping mitigating" should be "helping to mitigate".**

Response: Thank you for your correction. It was rephrased accordingly.

**Line 61: suggest removing "of".**

Response: Thank you for your suggestion. This was removed.

**Line 90: "extensive" is repeated.**

Response: Thank you for your suggestion. The sentence was corrected an now it reads: "...alongside extensive reference measurements, to generate the data for a comprehensive extensive in-depth performance assessment."

**Line 118: "inlfuenced" should be "influenced".**

Response: Thank you for your suggestion. Correction made.

**Line 128: "Quant" should be "QUANT".**

Response: Thank you for your suggestion. It was corrected.

**Line 142: "Polludrone: Poll" should be "Poll: Polludrone".**

Response: Thank you for your suggestion. The correction was made.

**Figure 2: Suggest using the same colors for the different sensors between the left and right panels.**

Response: Thank you for noticing this. This was corrected.

**Figure 3: Suggest moving this figure and associated discussion to the next section, since it is an assessment of performance against a reference rather than an assessment of inter-sensor consistency.**

Response: Thank you for your feedback regarding Figure 3. Although it assesses performance against a reference, it also reveals inter-device precision through the dispersion of points for sensors of the same brand. Its strategic location before the section "3.2 Device accuracy and collocation calibrations" provides a transition to discussions on accuracy, underscoring not only reference comparison but also variability among devices of the same make—essential for understanding sensor consistency and reliability. Thus, we would like to maintain Figure 3 in its current position, but will move it if the reviewer and editor insist. We have slightly adjusted the preceding text for clarity, as follows:

 In addition to showcasing inter-device precision, Fig. 3 also serves as a transition to accuracy evaluation (the focus of the subsequent section).

**Figure 8: These seems to be a switch between the use of uncalibrated and calibrated data between the left and right panels as well. It is not clear what these calibrations are based on, and the application of the calibration might be a contributing factor to the difference in performance, together with the move between sites. It may be more illustrative to present a comparison at both sites with either the calibrated or the uncalibrated data only.**

Response: We'd like to clarify that the "out-of-the-box" and "calibrated" data products are associated with specific periods (as summarised in Figures S2 and S3). Calibrations were performed by the companies using data from Manchester, during the first co-location period (Dec 2019 - Feb 2020). At the end of this period, the brands ceased providing "out-of-the-box" data and began supplying data adjusted for the co-location data from Manchester. A few days later, one quarter of the instruments were moved to London. Thus, while the data are labelled as "calibrated", it does not imply (in this case) that they have been corrected to local conditions in London.

Regarding the transition between uncalibrated and calibrated data across the panels, it's important to note that we lack access to the specific calibration methods used by the manufacturers. This limits our ability to comprehensively detail the foundation of these calibrations.

As for the observed differences in performance between sites, and the potential influence of the calibration approach employed, we acknowledge that both aspects can be significant. Although London and Manchester are classified as "urban background" sites, one might expect comparable sensor performance, disparate calibration methods—applied as manufacturers assimilate local reference data—may lead to divergent outcomes. This is exemplified by "Sensor A", which, upon relocation to London, exhibits a shift in bias while maintaining response linearity. In contrast, "Sensor B" shows a notable degradation in linearity. We suspect that the distinct calibration methodologies each company employs markedly influence these performance variances. Yet, beyond this speculation, the point we aim to highlight with this figure is the potential for end-users to implement simple corrections. Specifically, "Sensor A" appears amenable to linear correction, whereas for "Sensor B", such an approach may not yield significant benefits.

Concerning the suggestion to present a comparison at both sites using exclusively calibrated or uncalibrated data, we are limited by the nature of the data products available during the periods in question (as detailed in Figures S2 and S3).

The original text has been reworded in order to convey these points and now reads:

the response is notably noisier as the Standard Error (SE) — which is the dispersion of the data around the best-line fit line, i.e., the remaining error after bias correction. In scenarios akin to this latter, where there is a high variance in the residuals, a linear correction will not provide a significant improvement. While more sophisticated corrections could be applied, these will be limited by domain knowledge of the end-user, and potentially by other complex data sources that might be available. However, it is important to remember that additional post-processing could increase the risk of overfitting (Aula et al., 2022). On the other hand, for cases like the top plots, users might benefit from trying to correct them using simple linear correction (e.g. using reference instruments if available) or other approaches that could provide means for zero and span correction. A straightforward and cost-effective example could be the use of diffusion tubes for the case of NO2, as discussed in Section 3.6. The primary distinction between both systems' behaviour lies in the fact that the sensor located in the top row (Sensor A), even after being relocated to London, maintains a linear response (albeit slightly more degraded than that observed in Manchester, as indicated by the $R^2$ and RMSE). In contrast, Sensor B's response becomes significantly noisier upon relocation to London, as highlighted by the Standard Error (SE) —which represents the remaining error after applying a perfect bias correction. Despite both systems utilising identical sensing elements, the variance in residuals between them may stem from the distinct calibration approaches applied by the respective companies.

For cases resembling Sensor A, users might find it beneficial to implement simple linear correction methods (e.g., using reference instruments if available) or explore other strategies for zero and span correction. A practical and cost-effective approach, for example, is using diffusion tubes for $NO_2$ measurements, as discussed in Section 3.6. Conversely, in scenarios characterised by high variance in residuals, such as those observed with Sensor B, a-posteriori attempts to apply a simple linear correction are unlikely to result in significant improvement. While more sophisticated corrections are theoretically feasible, their effectiveness is limited by the end-user's domain knowledge and the availability of additional complex data sources. Furthermore, it is important to consider that excessive post-processing may lead to overfitting — a situation where a model excessively conforms to specific patterns in the training data, resulting in poor performance on new, unseen data (Aula et al., 2022).

**Lines 329-331: Sentence may be incomplete.**

Response: Thank you for noticing this. We have adapted the text. Please see the previous response text.

**Lines 373-374: The meaning of this is unclear; does this mean that results from 2 systems were combined (e.g., to increase coverage)? Or were coverage and REU assessed separately for each device and then data from both devices combined to create the density plots of Figure 10?**

Response: Thank you for your inquiry. Each sensor device was independently assessed in terms of Data Coverage (DC) and Relative Expanded Uncertainty (REU). After this, we aggregated the data to create the density plots for all units of a unique brand, thus illustrating the collective behaviour of NO2 sensors from the same company in relation to DC and REU. Recognizing that the original text may not have clearly conveyed this, we have revised it as follows:

 Figure 10 illustrates the collective behaviour of $NO_2$ sensors from each of the four companies with more than two working systems, showcasing their REU (y-axis) versus Data Coverage (DC, x-axis). Both parameters were calculated for each sensor system using a 40-day moving window approach and then aggregated by brand, ensuring a comprehensive analysis. This methodology leverages overlapping data from multiple sensors to provide a robust representation of company-wide sensor performance and aims to prevent biassed interpretations.

**Line 374: "systems, not" should be "systems not".**

Response: Thank you for your suggestion. The correction was made.

**Line 423-424: Please explain further the use of the reference data as a prior in this method.**

Response: Thank you for your comment. We have removed the mention of the prior as it distracted from the overall results, and instead have provided references to several papers that explain the general deweathering strategy.

**Lines 435-436: Consider changing one of "developments" or "developing".**

Response: Thank you for your comment. We have revised the sentence to eliminate the redundancy and improve the readability of the text. The modified sentence now reads:

 Advances are occurring rapidly, in both the measurement technology and particularly in the data post-processing and calibration.

**Lines 460-469: I would suggest adding a sentence earlier in the document (and perhaps in the abstract) noting that further analysis will be left for future publications. I was expecting at several points a more comprehensive presentation of results across all pollutants and for all phases of the study, while only particular aspects of the results were highlighted. This is alright, but I think it needs to be more clearly stated up-front that this is not a comprehensive presentation of the study results. I would also suggest, as a topic for future work, examining the manufacturer-suppled calibrations in more detail, seeing where these improved upon the raw and where they perhaps did not, and how robust the calibrations are to environment changes and movement of the sensors to new sites. This is briefly presented in several figures, e.g., Figure 8, but a more comprehensive assessment across all sensors and pollutants could be made.**

Response: We appreciate your feedback and have taken steps to clarify the scope and intent of our study both in the abstract and in the introduction of our document. In the abstract, we have added the following sentence:

While more comprehensive analyses are reserved for future detailed publications, the results shown here highlight the significant variation between systems, the incidence of corrections made by manufacturers, the effects of relocation to different environments, and the long-term behaviour of the systems.

Similarly, we have modified the introduction, and now reads:

This comprehensive approach offers unprecedented insights into the operational capabilities and limitations of these sensors in real-world conditions. Significantly, some of the insights gathered during QUANT have contributed to the development of the Publicly Available Specification (PAS 4023, 2023), which provides guidelines for the selection, deployment, maintenance, and quality assurance of air quality sensor systems. While this manuscript serves as an initial overview, detailed analyses of the measured pollutants and study phases, offering a more comprehensive perspective on sensor performance, are planned for future publications.

We appreciate your suggestion concerning the detailed examination of manufacturer-supplied calibrations. Indeed, this aspect is being considered in our current efforts and we anticipate publishing these findings in the near future.

**Supplemental Information, Lines 4-6: Indicate which of these channels and/or data products were considered for this study. Also report the sampling frequency for this sensor.**

Response: Thank you for your feedback on this point. We wish to clarify that the original supplementary text did indeed specify that the PA sensors provided data at a 2-minute resolution. Furthermore, in response to concerns about channels and data products, we have added a note to the manuscript for clarity, which states:

*Note: For this study, only Channel A and the data product "cf_atm" were included in the analysis and shown in the plots.

To ensure a comprehensive understanding of the sensor data utilised, we collected all data products offered by each company, preserving their native resolution. To enhance clarity, we have now included two additional tables in the supplementary materials—one for the QUANT study (table S4) and another for the WPS study (table S5). These tables summarise the data products collected during QUANT and their native resolution.

**Supplemental Information, Line 55: "y" should be "and".**

Response: Thank you for your suggestion. The correction was applied.

**Reviewer#3**

**GENERAL COMMENTS**

**This article provides an important contribution to the advancement of studies associated with air quality sensors. It provides a good overview and information about the QUANT study and some important results. Discussions associated with data quality add value in an important way to alert to errors and possible corrections associated with time and space. Shows the importance of using reference sensors in calibrations detailing correct use and necessary considerations.**

**I recommend this publication. However, as this is an important study that can be reused or used as a basis for others, I think it is important to go into more detail especially methodologically so that it can be continued and used as the authors suggest at the end.**

Response: Thank you for recognizing the contribution of our article and for your supportive remarks on the overview and insights provided by the QUANT study.

**SPECIFIC COMMENTS**

**Section 2.1**

**As spatial analyzes are carried out, I consider the spatial description of the areas of the article to be important such as distances and spatial layout. A spatial image would enhance spatial visualization and discussion. This arrangement is important in analyzing spatial differences and environmental conditions that influence the data.**

Response: Thank you for this suggestion. Recognizing this, we have expanded the description of the study areas to include more detailed information on distances and spatial layouts. Additionally, to further enhance spatial visualisation and support the discussion, we have incorporated satellite images taken from Google Maps into our manuscript.

**Section 2.3**

**Line 135. The sensors were implemented according to the manufacturer's specifications. Was any standardization found in the logistics or studied at this stage? I think it's important to describe this stage perhaps in supplementary material. The layout of the sensors, whether it was completely open or needed some protection, ground height, proximity to the reference, obstacles, necessary infrastructure, etc. These are all factors that influence the data and are still the subject of much discussion when it comes to implementing the sensors.**

Response: Thank you for highlighting this point. In response to your suggestions, we have adapted and renamed one of the sub-sections in the methodology, in order to describe these important points. Please refer to:

**2.3 Sensor deployment and data collection, co-located reference data and data products**

**Section about treatments, analysis, and metrics**

**It would be important to include a section describing the data analysis treatments and statistical metrics that were used for these specific analyses.**

Response: Thank you for your detailed suggestion. We've created a new section in the supplementary ("S5. Performance Metrics") that succinctly describe the statistical metrics employed in our analysis.

**It would be important to include: data standardizations such as sensor frequencies for comparison with the reference, if there was a change in frequency, how the amount of valid data for the calculations was considered; a description of the calibrations or validations applied; statistical metrics used in analyzes such as RMSE, REU, etc., a simple description would add a lot to the article; Another point would be the pollutants used (PM2.5, NO2) and because these if there are analyzes for the others, it would be interesting to mention.**

Response: Thank you for your detailed suggestion. Following previous reviewers' comments, we have taken steps to address these aspects in the manuscript:

-for the updated text on data standardisation and sensor frequencies, please refer to the "2.3 Data collection" section.

-in regards to the amount of valid data used for metrics and plots, please see the added text in the "Results" section.

-we've expanded our description on the calibration processes applied to the sensors in the newly created section "2.4 Data products and co-located reference data".

-as for statistical metrics, we've created a new section in the supplementary ("S5. Performance Metrics") that describes the statistical metrics employed in our analysis.

**SECTION 3**

**Why were some analyzes used PM2.5 and others NO2? Would there be any explanation?**

Response: Thank you for your inquiry. The primary air pollution issues in the UK are PM2.5 and NO2 exceedances, and as such in this overview paper we aimed to showcase the NO2 and PM2.5 measurements over those of other pollutants due to their relevance. The choice of our use of NO2 or PM2.5 for any particular example shown is either to highlight a specific facet of the data, such as the potential use of NO2 diffusion tubes to reduce NO2 sensor bias, or is arbitrary in order to avoid focussing more on one pollutant over another. We have added the following text to the results to clarify this:

The majority of examples presented here focus on $PM_{2.5}$ and $NO_2$ measurements, due to both a larger dataset available for these pollutants and their critical role in addressing the exceedances that predominantly impact UK air quality.

**From section 3.4 onwards, sensors are no longer specified from which manufacturer. For example, in Figure 7, which sensors are being compared? Is it just from one manufacturer? Or multiple manufacturers? Is only one sensor from each manufacturer considered or multiple?**

Response: Thank you for your question. To emphasise the broader implications and insights into sensor technology, we chose to anonymize figures illustrating brand-specific features. This aims to mitigate potential bias and foster a broader view of the technology performance, focusing on general trends rather than the performance of individual manufacturers. We've provided our reasoning in the "3. Results and discussion" section for clarity.

As for Figure 7, the original caption specifies that the comparison involves sensors from a single manufacturer, though we have anonymized the details to align with our overarching goal of emphasising generalizable findings.

**In figure 8, what would sensors A and B be, are they from the same manufacturer or different?**

Response: Thank you for your inquiry. Sensors A and B represent two distinct systems from different manufacturers. We have adapted the manuscript accordingly to clarify this point for our readers:

A second example of inter-location performance changing between locations is presented in Fig. 8, showing $NO_2$ data from two sensor systems (from two different manufacturers, identified as Systems A and B) (different brands, one shown on top of the other) before (left plots) and after (right plots) they were moved from Manchester to London in March 2020.

**Figure 10. Is the analysis for NO2? If yes, specify in the legend. The companies are unidentified, wouldn't it be possible to associate them with each one?**

Response: Thank you for your input. The specified corrections have been made. We've also identified the companies in Figure 10. As we responded to an earlier comment, we initially chose the anonymize companies to focus discussions on broad technological features over specific manufacturer data.

[revised manuscript text omitted]
. Situated in an urban background setting approximately four kilometres south of Manchester city center — the UK's second-largest metropolitan area with around 3.3 million residents — MAQS benefits from a strategic location on the University of Manchester's Fallowfield Campus. This location is notably distanced from direct traffic emissions, surrounded by student accommodations, university administrative buildings, and sports facilities. The campus's vicinity to shops, bars, and restaurants introduces a range of human activities, including varying levels of foot traffic and associated vehicular movement. Additionally, the presence of these commercial and recreational spaces, alongside residential buildings, contributes to the area's ambient air quality through emissions from heating and cooking, among other sources. For a visual representation of MAQS's surroundings, please refer to Figure S1 (panel a). The site experiences an average winter temperature of approximately 4-5°C with relative humidity around 87%, and an average summer temperature of about 16-17°C with relative humidity near 88%. Detailed information on MAQS's reference instrumentation and the methodologies employed for air quality measurements can be found in section S2. Data from MAQS are provided with a 1-minute time resolution, facilitating a granular temporal analysis of air quality metrics.

The London Air Quality Supersite (LAQS) is an urban background monitoring site located at Honor Oak Park (51° 26' 58.9"N 0° 02' 14.6"W) in Greater London, the third biggest European urban conglomeration with approx. 14.8 million inh. (avg. temp. in winter ~5 °C and RH ~84 %, avg. temp. in summer ~17 °C and RH ~72 %). All gas data provided by LAQS is 1-min time resolution and 15-min for PM.

The London Air Quality Supersite (LAQS, 51° 26' 58.9"N 0° 02' 14.6"W) serves as an urban background monitoring site, nestled within Honor Oak Park in Greater London. Situated 9 km southeast of the city center of the third-largest European urban conglomeration, LAQS offers a unique window into the air quality challenges of an area inhabited by approximately 14.8 million people. Nestled within the serene King's College sports grounds, is surrounded by middle-class neighbourhoods, abundant parks, and green spaces. This tranquil setting, is distanced from major roads and pollution sources, provides a representative snapshot of the ambient air quality typical of residential London. LAQS's surroundings are marked by a low level of commercial activity, with local shops and restaurants contributing minimally to the area's overall noise and bustle. Figure S1 (panel b) offers an aerial view of LAQS, illustrating the overall urban layout. The area is characterised by a temperate climate, experiencing average winter temperatures of around 5°C with RH of approx. 84%, and milder summers with temperatures averaging 17°C and RH of around 72%. Gas measurements at LAQS are conducted with a 1-minute time resolution, while PM data are collected at a 15-minute resolution (see section S2 for more details).

The  Fishergate roadside site (YoFi), located in the city of York (~210,000 inh., avg. temp. in winter of ~4°C and RH ~87 %, avg. temp. in summer around 15 °C and RH ~80 %). This site is a self-contained air quality monitoring station located very close to the city centre on a traffic island (53° 57' 06.9"N 1° 04' 33.1"W) surrounded by a residential/commercial area. This site was chosen to evaluate the LCS responses to a greater pollutant variability typical of traffic-related sites (in contrast with urban background monitoring stations as in the case of MAQS and LAQS). While PM and NOx data from YoFi are 1-hr time resolution, the O3 data is 1-min (deployed on the 15th of May 2020, specifically as part of the QUANT study).

The York Fishergate roadside site (YoFi, 53° 57' 06.9"N, 1° 04' 33.1"W), in the historic city of York, which is home to approximately 210,000 inhabitants (avg. temp. in winter of ~4°C and RH ~87 %, avg. temp. in summer around 15 °C and RH ~80 %). Situated just about 1 km from the city center on a traffic island, YoFi stands amidst a predominantly residential area that also encompasses commercial and light industrial elements. Unique to its location, the site is sandwiched between two lanes of Fishergate Road, a major avenue that bifurcates to facilitate traffic flow into and out of the city's southern part. Directly across from YoFi, a primary school adds to the daily human activity around the site, while the nearby River Ouse, located merely 300 metres to the west, contributes to the area's environmental characteristics. A vibrant commercial zone, featuring pubs and restaurants, is found just 100 metres to the north. Moreover, the site is flanked by Walmgate Stray, an expanse of recreational fields, located about 300 metres to the southeast, offering a green respite amidst the urban setting. Additional details can be visualised in Figure S1 (panel c), providing an aerial perspective of the site's key features and its urban context. This self-contained air quality monitoring station was specifically selected for the QUANT study to assess sensors' responses to the greater pollutant variability typical of traffic-related sites, contrasting with the urban background settings of MAQS and LAQS. YoFi provides data on PM and NOx with a 1-hour time resolution. Additionally, in a targeted effort to enhance our understanding of air quality dynamics, $O_3$ measurements (deployed on the 15th of May 2020, specifically as part of the QUANT study), utilising a 1-minute time resolution to offer detailed insights into temporal variations (refer to section S2 for more details).

[Figure]

[Figure]

[Figure]

**Figure S1: Aerial views of the air quality monitoring sites: a) MAQS, b) LAQS, and c) YoFi, captured from Google**
**Earth. These images illustrate the diverse urban settings of each site, emphasising aspects such as their proximity to**
**traffic sources, presence of green spaces, and the general urban layout. Image credits: Google Earth.**

**S2. Reference instrumentation, QA/QC, and data-sharing periods**

Table S1 summarises the reference instrumentation at each site, Table S2 describes some of the QA/QC processes
at the supersites, and Table S3 shows the data periods shared with the suppliers.

**Table S1. Research grade instrumentation used for the QUANT study.**

| Analyte | Manchester | London | York |
|---------|-----------|--------|------|
| **NO** | Thermo 42i-y (Chem) | Teledyne T200U (Chem) | Teledyne T200UP (Chem) |
| **NO$_2$** | *Teledyne T500U (CAPS) | *Teledyne T500U (CAPS) | |
| **O$_3$** | *Thermo 49i (UV) | *Teledyne 400E (UV) | *2B 205 (UV) |
| **PM** | *Palas FIDAS200 (OAS) | *Palas FIDAS200 (OAS) | *Met One BAM 1020 (BA) |

*Equivalent to reference (as defined in the European Air Quality Directive 2008/50/EC)

Acronyms: Chem: Chemiluminescence; CAPS: Cavity Attenuated Phase Shift Spectroscopy; UV: Ultraviolet; OAS: Optical aerosol spectrometer; BA: Beta attenuation.

**Table S2. Summary of Quality Assurance processes in MAQS and LAQS**

| Instrument | Frequency | *Process |
|------------|-----------|----------|
| NO$_y$ | At least monthly | Zero and span checks using standard cylinder and scrubber. Corrections to zero and span values. |
| NO$_2$ | Daily | Automatic zero and span checks using internal NO$_2$ diffusion tube and scrubber. Zero corrections, span monitored. |
| O$_3$ | Daily | Automatic zero and span checks using internal O$_3$ lamp and scrubber. Corrections to zero, span monitored. |
| CO | Every three hours & monthly | Zero checks every three hours and span checks monthly using onsite cylinder. Adjustments to zero and span values. |
| CO$_2$ and CH$_4$ | Regular | Stability checks using onsite cylinder, no corrections made. |
| *PM | Semiannual | Sizing response verified with Mono dust, flow rate checked with Gilibrator. |

*Checked with external standards by NPL every 6 months. These external standards are also used to provide a certification of the on-site standard cylinders. Final corrections to the data are provided by using the audit data to define the concentration of the on-site standards, with zero and span values interpolated between the calibration points.

**Sizing and flow checked every 6-month NPL audit process.1

     **Table S3. Reference data is shared with the sensor manufacturers.**

| | QUANT main study | | | Wider Participation Study | |
|---|---|---|---|---|---|
| **Reference dataset** | **Period** | **Released** | **Reference dataset** | **Period** | **Released** |
| **1** | 10-12-2019 - 17-02-2020 | 15-04-2020 | **1** | 17-06-2021 - 16-07-2021 | 23-07-2021 |
| **2** | 18-02-2020 - 17-08-2020 | 27-10-2020 | **2** | 01-12-2021 - 31-12-2021 | 26-01-2022 |
| **3** | 18-08-2020 - 17-02-2021 | 15-04-2021 | **3** | 01-05-2022 - 31-05-2022 | 15-06-2022 |

**S3. QUANT main study devices**

In this section, a brief description of the QUANT main study systems' components is offered.

PurpleAir (PA) (https://www2.purpleair.com) devices (PA-II-SD model, firmware v4.11) reports particulate
matter ($PM_1$, $PM_{2.5}$, and $PM_{10}$), and it was chosen for its penetration around the world. Two identical Plantower
PMS5003 (Plantower) sensors (channels A and B) are found in each PA. It offers two data products (2-min avg.
time): the "cf_atm" (for outdoor applications) and the "cf_1" (for indoor or controlled environment applications).
The PMS behaves like a nephelometer rather than an optical particle counter to measure the light scattered by the
PM (Ouimette et al., 2022) and is composed of a laser, a photodiode, a fan, and a microprocessor control unit.
They also measure temperature (Temp), relative humidity (RH), and atmospheric pressure (Pres) (Bosch). The
data can be communicated via Wi-Fi or stored locally (microSD card), which was the preferred way during the
colocation. No calibrated products are offered by the company.

*Note: For this study, only Channel A and the data product "cf_atm" were included in the analysis and shown in
the plots.

AQMesh (https://www.aqmesh.com) reports $NO_2$, NO, $O_3$ using electrochemical (EC) sensors (Alphasense), $CO_2$
with a non-dispersive infrared sensor (NDIR, Alphasense), $PM_1$, $PM_{2.5}$, and $PM_{10}$ through a light-scattering sensor
(Nephelometer, Environmental Instr.) with 1-minute time resolution (algorithm v5.1 for gases and v3.0 for PM).
This instrument also registers Temp, RH, and Pres (Solid-State sensors) (Zauli-Sajani et al., 2022) and the
sampling mechanism employs a pump. The collected data is sent to the company server via a cellular network and
post-processed (Temp, RH, and cross-interference correction) in the cloud by a proprietary algorithm. Finally, the
data is released to the final user via secure web login or through its Application Programming Interface (API).
Although the first 4 months of the deployment the data had a 15-min resolution, since then the provided resolution
is 1-min average.

AQY (v.1.0) is also a multi-species device (https://www.aeroqual.com) and measures $O_3$, $NO_2$, $PM_{2.5}$, $PM_{10}$,
Temp, and RH. This is the only device system that does not use Alphasense sensors for gases. While $O_3$ is
quantified using a metal oxide sensor ($WO_3$-based, Aeroqual Ltd), the $NO_2$ is measured by an EC sensor
(Membrapore type $O_3$/M5, Aeroqual Ltd) (Weissert et al., 2019). For PM it uses a light scattering method (Nova)

to convert size and particle count to a mass fraction and behaves like a nephelometer (Myklebust et al., 2022).
These LCS devices send their data (1-min time resolution) to the Aeroqual server via cellular (WiFi could also be
used for this purpose) or stored locally (microSD card). The non-local data access is through a web portal or via
API.

Zephyr units (https://www.earthsense.co.uk) measure PM (Nephelometer, Plantower), Temp & RH (Sensirion),
and Press (Bosch) (the sample uptake uses a fan). As most of the commercial units tested here, it used Alphasense
EC sensors (the "A series", a smaller version than the B series) for gases (NO, $NO_2$, and $O_3$). These devices send
their raw data to the server via a cellular network, where they pre-process the raw signals. We have secure access
to the measurements with a time resolution of 1-min per species through the website or via its API.

ARIsense v200 devices (https://quant-aq.com) measure NO, $NO_2$, $O_3$, CO (EC, Alphasense), $CO_2$ (NDIR,
Alphasense), Temp & RH (Sensirion), and Press (Bosch) (Cross et al., 2017). Of all the devices tested, this is the
only one that uses an Optical Particle Counter (OPC) for PM (Particles Plus). Communication is carried out
through a cellular network and the data products are accessed through a web portal or API (1-minute time
resolution). According to the company policy, only the gas data products are subjected to calibrations (if
colocation data is available).

**Table S4. Summary of sensor measurements and the time resolution data provided by participating companies in the Main**
**QUANT study.**

[revised manuscript text omitted]

**S5. Performance Metrics**

In the assessment of sensor measurement error, it is standard practice to employ a linear additive model, described by the following equation:

$$y_i = b_1 x_i + b_0 + \varepsilon_i \tag{1}$$

In this model, the dependent variable "y" represents the sensor measurements, while the independent variable "x" denotes the reference measurements. The coefficient $b_1$ corresponds to the slope of the regression line (the response sensitivity of the sensor relative to the reference) and $b_0$ is the ordinate at the origin (the sensor's output when the reference measurement is zero). $\varepsilon_i$, assumed to have a mean of zero and a standard deviation of $\sigma_\varepsilon$, captures the portion of "y" that cannot be explained by "x". For a sensor to perfectly match the reference measurements (i.e., y = x), $b_1$ would equal one, with both $b_0$ and $\varepsilon_i$ being zero.

*Coefficient of Determination ($R^2$)*

$R^2$ is an adimensional metric that quantifies the proportion of variance in the sensor measurements ("y") that can
be explained by its linear relationship with the reference measurements ("x"):

$$R^2 = \frac{\sum_{i=1}^{n} (x_i - \hat{y})^2}{\sum_{i=1}^{n} (y_i - \hat{y})^2}$$  (2)

As a bounded metric, $R^2$ varies between zero and one ($0 \leq R^2 \leq 1$), where a value closer
to one indicates a stronger linear association between the sensor and reference
data. Despite being one of the most widely used metrics in sensor evaluation, as
highlighted by Karagulian et al. (2019), $R^2$ comes with limitations that warrant careful consideration.
Notably, $R^2$ does not account for bias in the data; a regression line diverging from the ideal 1:1 relationship
between "x" and "y" does not affect its value. Additionally, $R^2$ is influenced by the dynamic range of the
measurements, which can skew its interpretation. Given these nuances, it is prudent to report $R^2$ alongside
complementary metrics that can offer a more rounded view of sensor performance. For a more in-depth analysis
of the limitations and proper use of $R^2$, readers are directed to the discussion in Legates and McCabe Jr. (1999).

***Mean Absolute Error (MAE) and Root Mean Squared Error (RMSE)***

MAE and RMSE (both dimensional metrics, expressed in the same units as the measured variable), also stand as
very popular metrics for performance evaluation, as they offer insights into the accuracy of sensors, presenting a
fuller picture than the $R^2$ alone. These metrics can be estimated as follows:

$$MAE = \frac{1}{n}\sum_{i=1}^{n} |y_i - x_i|$$  (3)

$$RMSE = \sqrt{\frac{1}{n}\sum_{i=1}^{n} (y_i - x_i)^2}$$  (4)

However, both MAE and RMSE quantify average errors. MAE does so by calculating the average magnitude of
errors without directionality, utilising absolute differences, while RMSE gauges the standard deviation of these
differences, highlighting the squared differences between sensor readings and reference grade measurements.
Although MAE and RMSE are both valued for their measure of accuracy, they bear distinct implications in
practice. MAE treats all errors equally, allocating proportional weight across the board. Conversely, RMSE
disproportionately penalises larger errors due to its squaring of difference values, an aspect noted by (Willmott
and Matsuura, 2005). This characteristic makes RMSE particularly sensitive to outliers, shaping its utility in
identifying and rectifying significant deviations.

***Mean Bias Error (MBE)***

The MBE quantifies the average bias in sensor measurements relative to reference values. Expressed in the same
units as the variable being measured, MBE reflects the systematic error, offering a straightforward indication of a
sensor's tendency to overestimate or underestimate the reference:

$$MBE = \frac{1}{n}\sum_{i=1}^{n} (y_i - x_i)$$  (5)

A zero value of MBE indicates no consistent over- or underestimation, while positive or negative values signal systematic bias in measurement. This simplicity in interpretation makes MBE particularly valuable for initial assessments of sensor accuracy and for guiding calibration efforts to correct for systematic bias. However, the MBE does not capture the precision of the measurements. For this reason, MBE is most effective when used in conjunction with other metrics, such as RMSE and MAE, to gain a comprehensive understanding of sensor performance, encompassing both systematic and random errors.

**Relative Expanded Uncertainty (REU)**

In contrast to single-value metrics such as $R^2$, RMSE, and MAE, which assess data sets as a whole, REU offers a "point by point" metric. This allows for graphical representations (like the REU in the concentration space or as a time series), offering detailed insights into measurement performance variability. The REU's mathematical framework is outlined in the "Guidance for the Demonstration of Equivalence of Ambient Air Monitoring Methods" (European Commission, 2010), as follows:

$$U(y_i) = \sqrt{\frac{RSS}{n-2} - u^2(x_i) + (y_i - b_0 - b_1 x_i)^2} \tag{6}$$

$$REU(y_i) = \frac{k.U(y_i)}{\hat{x}} \tag{7}$$

$$RSS = \sum_{i=1}^{n} (y_i - b_0 - b_1 x_i)^2 \tag{8}$$

here, $U(y_i)$ represents the measurement uncertainty [concentration units]; $REU(y_i)$ denotes the REU [percentage]; $u(x_i)$ is the random uncertainty of the reference monitor [concentration units]; "n" stand for the number of collocated data points considered; RSS is the Residual Sum of Squares; k is the coverage factor (set at 2 for a 95% confidence level).

A distinctive feature of REU is its incorporation of the uncertainty associated with the reference method (i.e., $u(x_i)$). This aspect recognizes that all measurements, including those from reference methods, are subject to inherent uncertainties. While calculating REU is more complex than traditional metrics, it's essential to acknowledge that, like any metric, REU is based on specific assumptions and considerations. These factors must be thoughtfully evaluated when interpreting data to ensure that conclusions are firmly rooted in the context of the study.

**Current guidance and normalisation efforts**

Table S6 summarises the key metrics addressed in some of the most recent guidance documents and technical standards. These metrics have been categorised under various labels: linearity, bias, error, uncertainty, data coverage, and inter-sensor precision. Each of these guidelines and regulations has its own set of procedures, protocols, and thresholds. Therefore, it is advisable for readers to consult the original documents for a detailed understanding of these specificities.

**Table S6. Summary of field evaluation metrics for sensors according to different guidelines and technical standards.**

| Feature | EPA[1&2] | CEN[3] | ASTM[4&5] |
| --- | --- | --- | --- |

| Pollutants covered | PM$_{2.5}$ & O$_3$ | NO$_2$, O$_3$, CO, SO$_2$ & Bencene | PM$_{2.5}$, PM$_{10}$ NO$_2$, O$_3$, CO & SO$_2$ |
|---|---|---|---|
| Linearity | R$^2$ | ---- | R$^2$ |
| Bias | Slope | Slope | Slope |
| | Intercept | Intercept | Intercept |
| Error | ---- | ---- | MAE |
| | RMSE | ---- | RMSE |
| | NRMSE | ---- | NRMSE |
| Uncertainty | ---- | REU | ---- |
| Data coverage | Data completeness | Data Capture | Data Capture Rate |
| Inter–sensor precision | SD | u$_{(bs,s)}$ | S$_{r,f}$ |
| | CV | ---- | ---- |

References in the table:

[1]EPA/600/R-20/279 Performance Testing Protocols, Metrics, and Target Values for Ozone Air Sensors.

[2]EPA/600/R-20/280 Performance Testing Protocols, Metrics, and Target Values for Fine Particulate Matter
Air Sensors.

[3]CEN/TS 17660-1: Air quality - Performance evaluation of air quality sensor systems - Part 1 Gaseous
pollutants in ambient air.

[4]ASTM D8406-22: Standard Practice for Performance Evaluation of Ambient Outdoor Air Quality Sensors
and Sensor-based Instruments for Portable and Fixed-point Measurement.

[5]ASTM WK74812: Standard Specification for Ambient Outdoor Air Quality Sensors and Sensor-based
Instruments for Portable and Fixed-Point Measurement.

Acronyms: EPA: U.S. Environmental Protection Agency; CEN: European Committee for Standardization;
ASTM: American Society for Testing and Material. CV: Coefficient of Variation; SD: Standard Deviation
(see the definition in the EPA Performance Testing Protocols); u$_{(bs,s)}$: Between sensor system uncertainty
(see the definition in the CEN TS 17660-1); S$_{r,f}$: field reproducibility standard deviation (see the definition
in the ASTM protocols).

**S6. Complementary plots**

[Figure]

**Figure S4. Inter-device precision of NO₂ measurements from "identical" devices across the 4 companies participating in QUANT is assessed using the "between sensor system uncertainty" metric (defined by the CEN/TS 17660-1:2021 as *u(bs, s)*). Each line represents this metric as a composite of all sensors per brand (excluding units with less than 75% data) within a 40-day sliding window.**

[Figure]

**Figure S5. The inter-device precision of O₂ measurements from "identical" devices across the 4 companies participating in QUANT is assessed using the "between sensor system uncertainty" metric (defined by the CEN/TS 17660-1:2021 as *u(bs, s)*). Each line represents this metric as a composite of all sensors per brand (excluding units with less than 75% data) within a 40-day sliding window.**

[Figure]

**Figure S6. Comparative analysis of "Sensor A" performance against two reference instruments for NO₂ measurements. The left plot shows the correlation with the Teledyne T500 (Cavity Attenuated Phase Shift Spectroscopy), while the right plot is against the Teledyne T200U (chemiluminescence) and specifically installed at the Manchester supersite for the QUANT study. The dashed red line represents the line of best fit for the sensor data against each reference, indicating a closer agreement with the T200U (slope=1.02) compared to the T500 (slope=0.73).**

[Figure]

**Figure S7. Comparative regression analysis and performance metrics of two distinct PM₂.₅ sensor systems benchmarked against a BAM for the top plots and a Fidas for the bottom plots. Each plot demonstrates the correlation and agreement between the sensor readings and the two equivalent-to-reference instruments in a roadside site located in York.**

**S7. NO₂ Diffusion tubes**

A diffusion tube co-location study was carried out between November 2020 and November 2021 at the MAQS, LAQS and York sites, using two types of diffusion tubes: the conventional (also known as LAQM, for Local Air Quality Management) and UUNN (for UK Urban NO2 Network). LAQM tubes have an open end and capture NO₂ which is converted to nitrite when reacting with triethanolamine (TEA) for subsequent analysis. On the other hand, UUNN tubes, similar in the sampling process to LAQM, include an amorphous polyethylene filter at the open end to further mitigate the effect of wind on NO₂ measurements. For more details refer to (Butterfield et al., 2021). Both types of tubes (conventional and UUNN) were installed in duplicates, either in shelters (to limit the incidence of wind) or directly exposed without protection in mounting blocks. Figure S5 illustrates the performance comparison of traditional diffusion tubes and a sensor system in Manchester. The data from these
diffusion tubes have been used to correct the sensor shown here and explained in detail in Section 3.6 (Figures 9b
and 9c).

[Figure]

**Figure S8. The left plot displays the correlation between an air quality sensor's readings and those from a reference**
**monitor for NO₂, while the right plot demonstrates the LAQM diffusion tube performance. The LAQM plot shows**
**a tighter correlation with the 1:1 line, indicating a higher accuracy in measuring NO₂ concentrations for the period**
**Nov 2020 - Nov 2021 at the Manchester supersite (blue dots represent monthly averages).**